# Lifestyles shape genome size and gene content in fungal pathogens

Anna Fijarczyk[1,2,3,4]*, Pauline Hessenauer[2,5], Richard Hamelin[2,5,6], Christian R Landry[2,3,4,7]

[1]Département de biologie, Université Laval, Québec, Canada; [2]Institut de Biologie Intégrative et des Systèmes (IBIS), Université Laval, Québec, Canada; [3]PROTEO, Le réseau québécois de recherche sur la fonction, la structure et l'ingénierie des protéines, Université Laval, Québec, Canada; [4]Centre de Recherche en Données Massives (CRDM), Université Laval, Québec, Canada; [5]Département des sciences du bois et de la forêt, Université Laval, Québec, Canada; [6]Department of Forest and Conservation Sciences, The University of British Columbia, Vancouver, Canada; [7]Département de biochimie, microbiologie et bio-informatique, Université Laval, Québec, Canada

*For correspondence:
anna.fijarczyk@uqo.ca

Competing interest: The authors declare that no competing interests exist.

## eLife Assessment

This **important** study addresses a topic that is frequently discussed in the literature but is under-assessed, namely correlations among genome size, repeat content, and pathogenicity in fungi. Contrary to previous assertions, the authors found that repeat content is not associated with pathogenicity. Rather, pathogenic lifestyle was found to be better explained by the number of protein-coding genes, with other genomic features associated with insect association status. The results are considered **solid**, although there remain concerns about potential biases stemming from the underlying data quality of the analyzed genomes.

**Abstract** Fungi display a wide range of lifestyles and hosts. We still know little about the impact of lifestyles, including pathogenicity, on their genome architecture. Here, we combined and annotated 552 fungal genomes from the class Sordariomycetes and examined the association between 13 genomic and two lifestyle traits: pathogenicity and insect-association. We found that pathogens, on average, tend to have a larger number of protein-coding genes, including effectors and tRNA genes. In addition, the non-repetitive size of their genomes is larger than that of non-pathogenic species. However, this pattern is not consistent across all groups. Insect endoparasites and symbionts have smaller genome sizes and genes with longer exons; moreover, insect-vectored pathogens possess fewer genes compared to those not transmitted by insects. Our study shows that genes are the main contributors to genome size variation in Sordariomycetes and that pathogens can exhibit distinct genome architectures, depending on their host and vector interactions.

## Introduction

The growing emergence of severe infectious diseases in humans, crops, and wildlife caused by fungi prompts strengthened efforts to uncover the molecular mechanisms of pathogenic traits. This requires understanding the forces shaping the evolution of fungal pathogen genomes (*Fisher et al., 2020*; *Rokas, 2022*). Fungal genome sizes span from 3 Mb in microsporidians (*Biderre et al., 1997*) up to 892 Mb reported in a rust fungus (*Mohanta and Bae, 2015*; *Tavares et al., 2014*), and fungal

pathogens score both among the smallest and the largest of them. Effectors, including small secreted proteins and RNAs, are fundamental in host colonization and defense against the host immune system and are also known to play a key role as virulence factors, i.e., molecular mechanisms responsible for causing damage to the host (*Stergiopoulos and de Wit, 2009*). Other genomic characteristics or processes correlated with or implicated in the evolution of pathogens include the presence of specialized gene families (*Raffa and Keller, 2019*), horizontal gene transfer (*Sahu et al., 2023*; *McDonald et al., 2019*), copy number variation of pathogenicity-relevant genes (*Bergin et al., 2022*), expansion of transposable elements (*Oggenfuss and Croll, 2023*), and genome compartmentalization (*Raffaele and Kamoun, 2012*; *Wacker et al., 2023*). Yet we still have limited knowledge of how important and frequent different genomic processes are in the evolution of pathogenicity across phylogenetically distinct groups of fungi and whether we can use genomic signatures left by some of these processes as predictors of pathogenic state.

Fungal pathogen genomes—particularly those of plant pathogens—are often characterized by large sizes, with expansions of TEs, and a unique presence of a compartmentalized genome with fast and slow evolving regions or chromosomes (*Raffaele and Kamoun, 2012*; *Möller and Stukenbrock, 2017*). Such accessory genomic compartments could facilitate the fast evolution of effectors (*Dong et al., 2015*). Similarly, TE expansions have been implicated in the evolution of new virulence genes in emerging pathogens (*Bao et al., 2017*; *Wacker et al., 2023*). Even though such architecture can facilitate pathogen evolution, it is currently recognized that its origin is more likely a side effect of a species evolutionary history rather than being caused by pathogenicity (*Torres et al., 2020*). Effectors are considered pivotal for pathogen evolution, along with the diversification of other gene families (*Muszewska et al., 2011*; *Baroncelli et al., 2016*; *Sipos et al., 2017*). As the number of genes is strongly correlated with fungal genome size (*Stajich, 2017*), such expansions could be a major contributor to fungal genome size. Notably, not all pathogenic species experience genome or gene expansions or show compartmentalized genome architecture. While gene family expansions are important for some pathogens, the contrary can be observed in others, such as Microsporidia. Due to the transition to an obligatory intracellular lifestyle, these fungi show signatures of strong genome contractions and reduced gene repertoire (*Katinka et al., 2001*) without compromising their ability to induce disease in the host. This raises questions about universal genomic mechanisms of transition to pathogenic state.

The size of genomes is dictated by the balance of mutation, selection, and genetic drift. Neutral evolution hypotheses generate predictions regarding the size of the genomes, the size of their deleterious non-coding parts, the number of genes, and the mutation rate (*Lynch and Conery, 2003*; *Petrov, 2002*; *Lynch, 2007*). In eukaryotes, the efficiency of purifying selection depends on the size of the species' effective population size ($N_e$) and will thus determine the rate of loss of slightly deleterious non-coding DNA, including mobile genetic elements or introns, leading to larger genomes with more repetitive content in species with smaller $N_e$. In bacteria, bias towards deletion rates will lead to shortening of the genome in small $N_e$ (*Mira et al., 2001*). These hypotheses also explain distinct genome evolution in species experiencing $N_e$ bottlenecks due to reproductive and lifestyle strategies, such as endosymbionts, endoparasites, or species with specialized vectors (*Mira and Moran, 2002*).

In ascomycetes, the evolution of genome size on a broad phylogenetic scale supports the neutral hypothesis of genome evolution, but in different ways. In species with larger genomes, indicators of drift, such as increased intron frequency and decreased gene density are associated with genome expansions, but in species with small and medium-sized genomes, those indicators are associated with genome size reductions (*Kelkar and Ochman, 2012*). Other phylogenomic studies investigating a wide range of Ascomycete species, while not explicitly focusing on the neutral evolution hypothesis, have found strong phylogenetic signals in genome evolution, reflected in distinct genomic traits (e.g. genome size, gene number, intron number, repeat content) across lineages or families (*Shen et al., 2020*; *Hensen et al., 2023*). Variation in genome size has been shown to correlate with the activity of the repeat-induced point mutation (RIP) mechanism (*Hensen et al., 2023*; *Badet and Croll, 2025*), by which repeated DNA is targeted and mutated. RIP can potentially lead to a slower rate of emergence of new genes via duplication (*Galagan et al., 2003*), and hinder TE proliferation, limiting genome size expansion (*Badet and Croll, 2025*). Variation in genome dynamics across lineages has also been suggested to result from environmental context and lifestyle strategies (*Shen et al., 2020*), with Saccharomycotina yeast fungi showing reductive genome evolution and Pezizomycotina filamentous

fungi exhibiting frequent gene family expansions. Given the strong impact of phylogenetic membership, demographic history ($N_e$), and host-specific adaptations of pathogens on their genomes, we reasoned that further examination of genomic sequences in groups of species with various lifestyles can generate predictions regarding the architecture of pathogenic genomes.

In this study, we investigate how fungal genome size and complexity associates with pathogenic lifestyle in the diverse ascomycete class of Sordariomycetes fungi. Sordariomycetes have rich genomic resources and include species with a wide range of lifestyles. Apart from plant pathogens and saprotrophs, this class contains several groups of insect endoparasites and species vectored by insects. Because endoparasitism and presence of vectors can impact genome evolution, we considered two non-exclusive lifestyle traits: pathogenicity and insect association. We studied 552 genomes, including 14 newly sequenced and assembled genomes and their associated 13 genomic traits (genome size, genome size excluding repeats, the number of genes, repeat content, GC content, number of introns, intron length, exon length, the fraction of genes with introns, intergenic length, the number of tRNA and pseudo tRNA genes) to answer the following questions: (1) which genomic traits predict genome size, (2) does larger genome size correlate with pathogenic lifestyle, and (3) are there genomic traits that are shared across all pathogens.

## Results

### Genome assemblies and phylogeny of Sordariomycetes

We analyzed 552 genomes from the class Sordariomycetes (Ascomycota) together comprising fungi mostly represented by plant pathogens, saprotrophs, and entomopathogens (*Supplementary file 1A*). Most genomes were downloaded from NCBI or other sources (*Supplementary file 1A*) and complemented with 14 genomes sequenced and assembled in this study (*Supplementary file 1B*). Genome size ranged from 20.7 Mbp in *Ceratocystiopsis brevicomis* (CBS 137839) to 110.9 Mbp in *Ophiocordyceps sinensis* (IOZ07) and the number of *ab initio* gene models ranged from 6280 in *Ambrosiella xylebori* (CBS 110.61) to 17,878 in *Fusarium langsethiae* (Fe2391). An overview of genome assemblies and their annotations, in relation to different sequencing technologies and other genomic databases, is provided in Appendix 1, *Appendix 1—table 1* and *Appendix 1—figures 1–7*. A maximum likelihood tree was generated from 1000 concatenated single-copy conserved proteins and had a 100% bootstrap support for all major nodes except for one (support of 82% for the split between two subclades of Ophiostomatales, *Figure 1A*).

### Genome size predictors

We examined the correlation between genome size (bp) and 11 genomic traits, including the number of genes, the repeats proportion in genome assembly, the size of the assembly excluding repeat content (bp), GC content, the mean number of introns per gene, mean intron length (bp), mean exon length (bp), the fraction of genes with introns, mean intergenic length (bp), and the number of tRNA and pseudo tRNA genes.

Most genomic traits were correlated with genome size and with each other, with the strongest positive correlation observed between the genome size, the genome size excluding repeats, and the number of genes (*Figure 1B*). Exon length, number of genes with introns, and GC content were negatively correlated with genome size. We extracted three major principal components from all 11 traits, which together explained 66% of variance among species. The first PC was mostly driven by coding traits (exon length, number of introns, genome without repeats, genes), whereas the second PC was mostly driven by non-coding parts of the genome (repeats, intergenic length, GC, and intron length, *Figure 1—figure supplement 1*). The third PC was mainly driven by the proportion of genes with introns, tRNAs, and number of introns. Modeling genome size using these three PCs showed that all axes were important predictors of genome size, with PC1 being the strongest (*Figure 1C*). PC1 also showed negative interaction with PC2 and PC3. This suggests that fungi with large genomes have, in general, either an excess of non-coding elements or an excess of coding elements but rarely both of them at the same time.

We calculated the genome-wide ratio of non-synonymous to synonymous substitution rates (dN/dS) as a proxy for an effective population size ($N_e$) (*Kelkar and Ochman, 2012*; *Lefébure et al., 2017*) to determine its association with the non-coding portion of the genome. We found no correlation

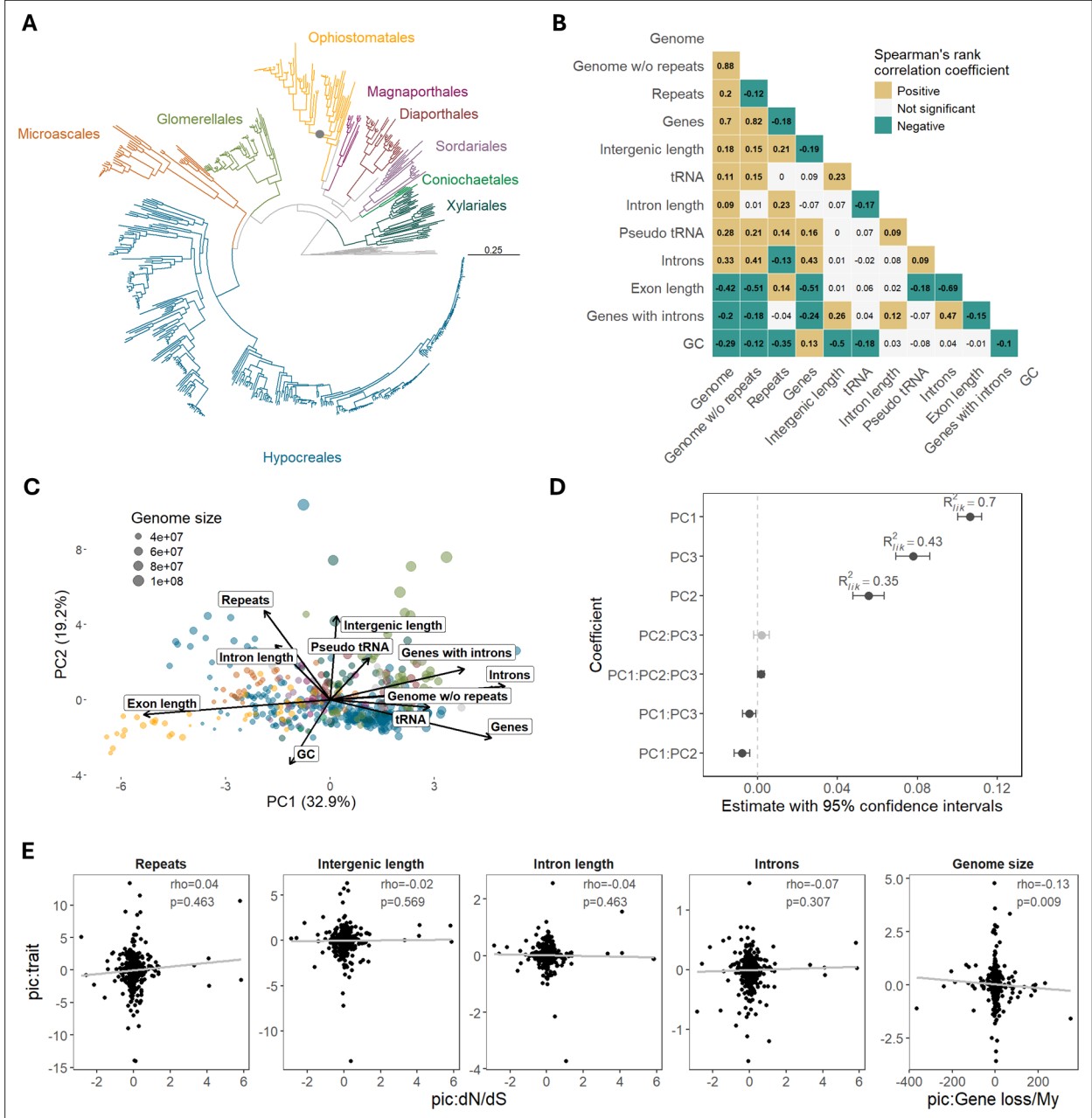

**Figure 1.** Several genomic traits are correlated with genome size in Sordariomycetes. (**A**) Maximum likelihood tree based on 1000 concatenated protein sequence alignments calculated with IQ-TREE using ultrafast bootstrap approximation (n=563 species). The largest orders are indicated with different colors. Bootstrap support for all major clades except one within Ophiostomatales (82%, black dot) reached 100%. (**B**) Spearman's rank correlation for phylogenetic independent contrasts of all pairwise combinations of genomic traits. (**C**) Principal component analysis of genomic traits. Colors correspond to orders depicted in A. (**D**) Model of genome size. On the y axis are the model coefficients with 95% confidence intervals obtained with phylogenetic generalized least squares model (PGLS) fitted to 555 genomes. Principal components correspond to the three main principal components based on 11 genomic traits. (**E**) Testing hypotheses of genome size evolution. The first four plots show the correlation of dN/dS as a proxy of $N_e$ with the non-coding elements of the genome. The last plot is a correlation of gene loss rate with genome size. Correlations were tested using Spearman's rank correlation on phylogenetic independent contrasts. Loadings of each principal component from panel C are shown in *Figure 1—figure supplement 1*. Data underlying figures are in *Figure 1—source data 1–4*.

The online version of this article includes the following source data and figure supplement(s) for figure 1:

**Source data 1.** Spearman's rank correlation coefficients for pairs of genomic traits.

**Source data 2.** Eigenvectors calculated all genomic traits.

**Source data 3.** Loadings calculated on all genomic traits.

*Figure 1 continued on next page*

*Figure 1 continued*

**Source data 4.** Contrasts calculated for genomic traits.

**Figure supplement 1.** Loadings of the main PCs.

between dN/dS and non-coding traits, such as repeat content, intergenic length, intron length, or number of introns (*Figure 1D*); therefore, we found no confirmation of accumulation of slightly dele-terious DNA in species that have high dN/dS indicative of small $N_e$. As the amount of non-coding elements is generally minimal in our dataset compared to other eukaryotes, such correlation may be difficult to detect. The rate of gene loss was negatively correlated with genome size (*Figure 1D*) but was not correlated with dN/dS (Spearman's rho = −0.05, *p*-value=0.44). This suggests that variations in deletion rates may influence genome size dynamics in this group of fungi.

## Genomic traits associated with pathogenicity

We found that pathogenic fungi did not have larger genomes, but instead had greater total number of genes, effectors, tRNAs, and fewer non-coding regions (*Figure 2B*). However, these patterns were not consistent across the phylogeny (*Supplementary file 1B*). To explore genomic traits associated with pathogenicity, we applied three methods that account for phylogenetic non-independence: an MCMC evolutionary model of binary traits (BayesTraits), phylogenetic logistic regression (Phyloglm), and Random Forest analysis. We assessed the same 11 genomic traits as before, adding genome size and the number of effectors. The distributions of genomic traits across pathogenic and non-pathogenic species are shown in *Figure 2—figure supplement 1*. All three methods consistently pointed to the number of effectors and tRNA genes as predictors of pathogenicity (*Figure 2B*, *Figure 2—figure supplement 2*). There was no support for a positive association between repeats and pathogenicity, and only one method indicated that pathogens tend to have larger genome sizes (*Figure 2B*). The total number of genes, genome size without repeats, and the number of pseudo-tRNAs were also positively associated with pathogenicity in at least two methods.

Our dataset has more pathogenic species than non-pathogenic ones (1.8:1), and this is true in particular for species with medium and large genomes (>50 Mb, 4.5:1). This could represent a genuine pattern or a bias due to the focus on sequencing pathogenic species with larger genomes. Even though we didn't find an association between pathogenicity and genome size, we accounted for potential sampling bias to test the robustness of our findings. Species were grouped by genome size, and we subsampled equal numbers of pathogenic and non-pathogenic species from each group, repeating the process 10 times. These 10 subsets were analyzed using the same methods in parallel, and nearly all confirmed that the number of effectors and tRNA genes were the strongest predictors of pathogenicity (*Figure 2C*). Genome size analyses yielded mixed results. Out of the 10 random subsets, BayesTraits identified co-evolution of genome size with pathogenicity but with no clear direc-tional trend. In contrast, Phyloglm coefficient estimates for genome size varied, showing either posi-tive or negative associations depending on the subset. The analysis also revealed some evidence of a negative association between pathogenicity and non-coding elements, such as repeats, intron length, number of introns, and intergenic regions. This could explain the absence of consistent posi-tive correlation between pathogenicity and genome size, as the predicted increase in gene number in pathogens is likely offset by a reduction in non-coding elements.

We also examined whether associations with pathogenicity are consistent across the entire phylo-genetic tree or vary between branches. The model with varying rates of evolution provided a signifi-cantly better fit for all genomic traits (*Supplementary file 1C*).

## Genomic traits of insect-associated (IA) species

Our analyses revealed that IA species tend to have smaller genomes and genes with longer exons on average (*Figure 3A*, *Figure 2—figure supplement 1*). All three methods consistently showed a positive association between exon length and IA species (*Figure 3A*). Additionally, there was support for a negative association between IA species and genome size, genome size excluding repeats, and gene number. Similar to pathogenicity, the evolution of IA traits varied across the phylogenetic tree, as indicated by the covarion models (*Supplementary file 1D*).

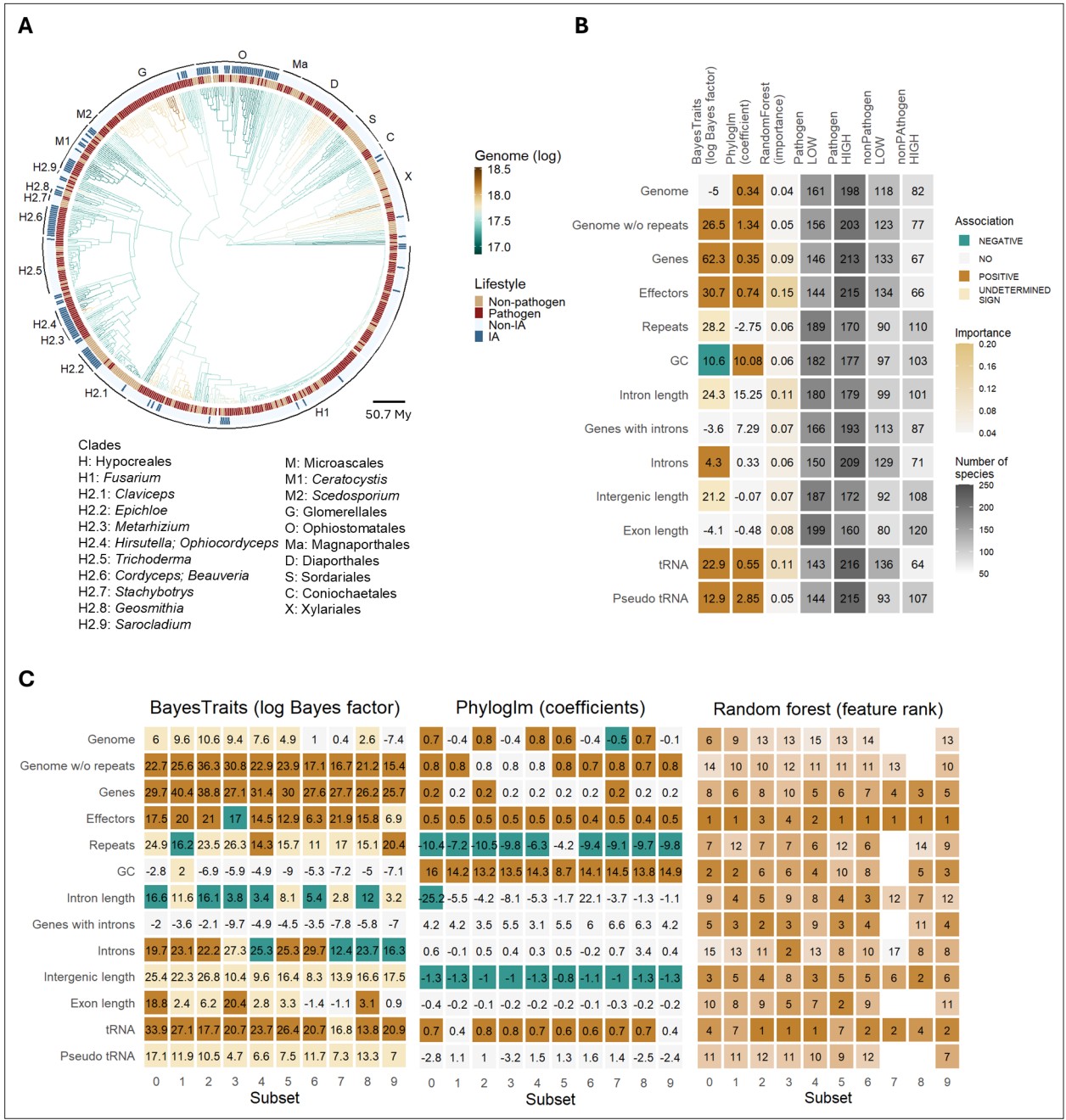

**Figure 2.** Genomic traits associated with pathogenicity. (**A**) Time-scaled phylogeny of Sordariomycetes colored by the reconstructed genome size displayed in log scale. The outside heatmap indicates species which are pathogenic and insect-associated (IA). Letters and numbers correspond to clades used in subsequent analyses, with representative genera from each clade listed in the legend. The scale bar indicates the branch length in millions of years (My). (**B**) Association of 13 genomic traits with pathogenicity estimated with BayesTraits, phylogenetic logistic regression (Phyloglm), and Random Forest classifier. In BayesTraits and Phyloglm, colors correspond to statistically significant associations, either positive (brown) or negative (green). In Phyloglm, these are based on coefficient sign; in BayesTraits, they were inferred based on transition rates between binary traits estimated from the model. 'YES' indicates that dependent co-evolution was detected with BayesTraits but a uniform direction of association could not be deduced from transition rates. Gray color scale depicts prevalence of species classified as pathogenic and non-pathogenic with traits size below median (LOW) or above median (HIGH). (**C**) Same three methods repeated for 10 subsets of the data with an equal number of pathogens and non-pathogens from each genome size bin. In Random Forest analysis, rank of the trait, instead of importance is given. Distributions of genomic traits are shown in *Figure 2—figure supplement 1*. Transition rates (modeled gains and losses of traits) are shown in *Figure 2—figure supplement 2*. Data underlying figures are in *Figure 2—source data 1–4*.

The online version of this article includes the following source data and figure supplement(s) for figure 2:

*Figure 2 continued on next page*

*Figure 2 continued*

**Source data 1.** Results of three tests of association of genomic traits with pathogenicity.

**Source data 2.** Counts of species classified as pathogenic and non-pathogenic with traits of a given size class.

**Source data 3.** Results of three tests of association of genomic traits with pathogenicity conducted in random balanced subsets.

**Source data 4.** Counts of gains and losses among four transition types obtained in BayesTraits run.

**Figure supplement 1.** Distributions of genomic traits across lifestyles.

**Figure supplement 2.** Numbers of gains and losses among four transition types obtained in BayesTraits run.

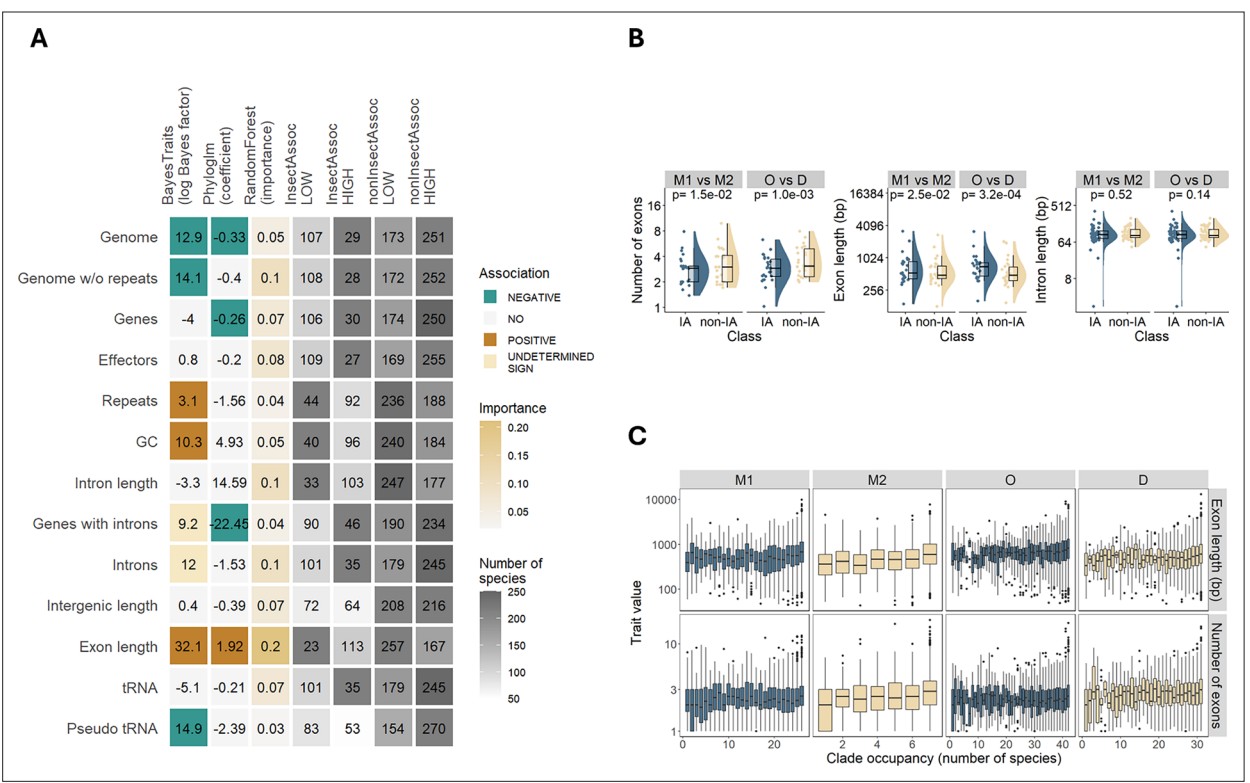

**Figure 3.** Genomic traits associated with insect-association (IA). (**A**) Association of 13 genomic traits with IA estimated with BayesTraits, phylogenetic logistic regression (Phyloglm), and Random Forest classifier. In BayesTraits and Phyloglm, colors correspond to statistically significant associations, either positive (brown) or negative (green). In Phyloglm, these are based on coefficient sign, in BayesTraits they were inferred based on transition rates between binary traits estimated from the model. 'YES' indicates that dependent co-evolution was detected with BayesTraits but a uniform direction of association could not be deduced from transition rates. Gray color scale depicts prevalence of species classified as IA and non-IA with traits size below median (LOW) or above median (HIGH). (**B**) Comparison of exon and intron metrics in 38 one-to-one orthologs between two clades, Microascales (M), Ophiostomatales (O), and Diaporthales (D). In the comparisons, one taxon is IA (M1, O) and another one is non-IA (M2, D). Exon/intron metrics were averaged within each clade and compared using a paired Mann-Whitney U test. (**C**) Correlation between prevalence of gene families within clades and two exon metrics, in the same four clades. In all clades, more common gene families have longer and more exons (negative binomial generalized linear model, *p-values* <0.05). The sample sizes per each bin (number of gene families) varied between 8 and 3148 in clade O, 12 and 4158 in clade D, 19 and 3764 in clade M1, 110 and 4740 in clade M2. Box plots in B and C indicate 25th (first quartile) and 75th percentiles (third quartile) of distributions, with median values in between. The whiskers indicate the maximum and minimum values within the range measured between the quartile and 1.5 times the interquartile range. Data underlying figures are in *Figure 3—source data 1–4*.

The online version of this article includes the following source data for figure 3:

**Source data 1.** Results of three tests of association of genomic traits with insect-association.

**Source data 2.** Counts of species classified as insect-associated and non-insect-associated with traits of a given size class.

**Source data 3.** Exon and intron metrics in 38 one-to-one orthologs.

**Source data 4.** Prevalence of gene families within clades and two exon metrics, number, and length of exons.

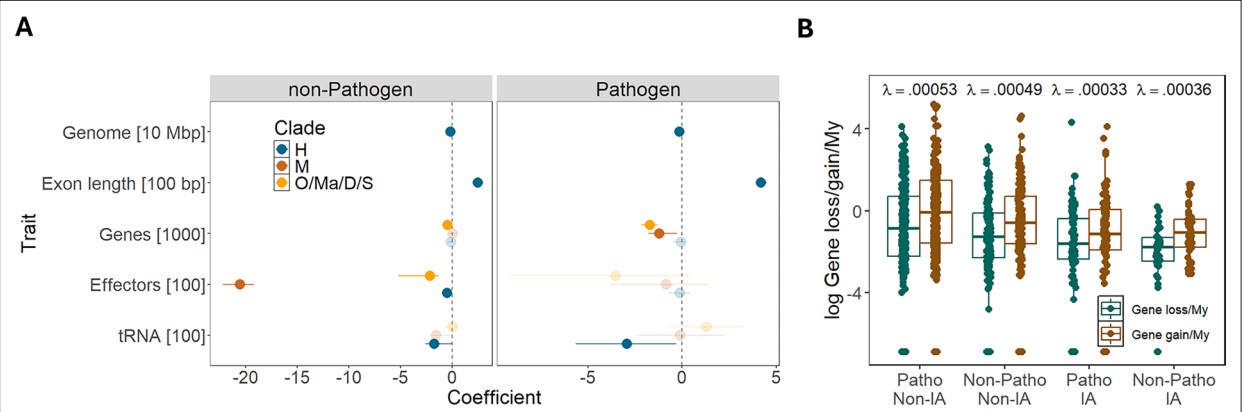

**Figure 4.** Evolution of pathogens with different lifestyles. (**A**) Models of insect-associated (IA) trait fitted with phylogenetic logistic regression using each of the five genomic traits and pathogenicity as predictors. Dots show coefficients of genomic traits for pathogenic and non-pathogenic species with 95% credible intervals and non-transparent colors indicating values with credible intervals not overlapping zero. Number of species used in each model were 317 for clade H (216 pathogenic and 101 non-pathogenic species), 33 for clade M (25 pathogenic and 8 non-pathogenic species), and 106 for clade O/Ma/D/S (53 pathogenic and 53 non-pathogenic species). (**B**) Rates of gene loss and gain in four groups of species with different pathogenicity and IA status, estimated based on 527 small gene families using birth and death model in the program CAFE v5. Estimates of gene evolutionary rates ($\lambda$) are shown above the box plots. Sample sizes were 267 non-IA pathogens, 147 non-IA non-pathogens, 88 IA pathogens, and 46 IA non-pathogens. Box plots indicate 25th (first quartile) and 75th percentiles (third quartile) of distributions, with median values in between. The whiskers indicate the maximum and minimum values within the range measured between the quartile and 1.5 times the interquartile range. Data underlying figures are in *Figure 4—source data 1 and 2*.

The online version of this article includes the following source data for figure 4:

**Source data 1.** Coefficients of genomic traits for pathogenic and non-pathogenic species with credible intervals estimated with phylogenetic logistic regression.

**Source data 2.** Rates of gene loss and gain in four groups of species with different pathogenicity and insect-associated (IA) status.

To ensure the association with exon length was not driven by a specific group of genes or species, we compared exon length and number in one-to-one orthologs between IA and non-IA species from two phylogenetic groups. In both phylogenetic groups, IA species had longer and fewer exons compared to non-IA species, while intron length was not different (*Figure 3B*). We also tested whether genes with longer exons were more likely to be retained in IA species compared to non-IA species. While genes present in most species within a clade (i.e. genes less likely to be deleted) were enriched for longer exons, this pattern was observed in both IA and non-IA species (*Figure 3C*).

## Pathogens gene evolution depends on lifestyle

Given that our results indicated genome size and gene content vary depending on fungal lifestyle, we investigated how IA pathogens evolve compared to non-IA ones. Using phylogenetic logistic regression, we modeled the IA trait, incorporating key genomic traits one at a time as predictors and including pathogenicity as an interaction term. This analysis was performed separately in three distinct fungal clades (H, M, and a O/Ma/D/S comprising four subclades, *Figure 2A*). Models incorporating genome size and exon length could only be fitted for clade H, revealing a negative association between IA species and genome size, and a positive association with exon length in both pathogens and non-pathogens (*Figure 4A*). In clades O/Ma/D/S and M, genes showed a stronger negative association with IA in pathogens compared to non-pathogens. In clade H, tRNAs exhibited a stronger negative association with IA in pathogens than in non-pathogens. Additionally, effectors were negatively associated with IA in non-pathogens across all three clades, but not in pathogens.

Host-pathogen interactions often result in rapid gene turnover due to ongoing adaptation to host defense mechanisms, leading to gene family expansions and contractions (*Gómez Luciano et al., 2019*; *Pendleton et al., 2014*; *Hartmann and Croll, 2017*; *Stalder et al., 2023*). To explore this, we analyzed a subset of ~500 gene families to test whether the rate of gene evolution ($\lambda$, the rate of gene loss and duplication *Mendes et al., 2021*) differs across species with different lifestyles. Species were categorized into four groups based on their pathogenicity and IA status, and we fitted models

with one, two, or four rate parameters. The model with four distinct rates was significantly better than the simpler models (likelihood ratio tests (LRT) 4 vs 1 rates: $\chi^2$=7.8, $p$-value=0; LRT 4 vs 2 rates: $\chi^2$=6, $p$-value=0.002). Gene evolutionary rates were higher in non-IA species and lower in IA species, regardless of pathogenicity (*Figure 4B*). Non-IA pathogens showed the highest evolutionary rates, while IA pathogens exhibited the lowest.

These results indicate that IA species are associated with slower gene evolutionary rates overall, and some IA clades are also associated with fewer genes, particularly in pathogenic species. However, some genes important for pathogens, such as effectors, are not depleted to the same extent as in non-pathogens.

## IA plant pathogens preferentially lose genes important for host colonization

To investigate the functional gene classes gained or lost in fungi with different lifestyles, we calculated the fold change in the number of various gene classes across 19 clades (*Figure 2A*) since their last common ancestor with sister clades. The clades include plant pathogenic fungi (H1, Ma, G, D), entomopathogens (H2.3, H2.4, H2.6), insect-vectored species (M1, O, H2.8, H2.2), saprotrophs (S), plant symbionts (H2.1, H2.2), and mixed lifestyle groups (*Figure 5A*), which display variation in the number of genes and genome size (*Figure 5B*). We utilized a range of databases to annotate genome assemblies or *ab initio* gene models (as detailed in the Methods).

Nearly all IA clades (M1, H2.8, O, H2.4, H2.2), along with a non-IA clade derived from an IA ancestor (H2.1, IA status of the MCRA of H2.1, H2.2, H2.3, and H2.4 estimated with BayesTraits equal to 0.97 [0.96–0.99]), a saprotrophic clade (S), and one mixed clade (H2.9), experienced the largest gene losses (*Figure 5C*). Entomopathogens exhibited a range of changes, with H2.4 showing mostly gene losses, H2.3 mostly gene gains, and H2.6 showing a mix of both. Conversely, the three clades that experienced the most gene gains were plant pathogenic clades (D, H1, G) and one IA clade (H2.3).

Insect-vectored species (M1, O, H2.8) demonstrated the highest gene losses in genes associated with secondary metabolite pathways, defense mechanisms, carbohydrate and lipid transport and metabolism (*Figure 5C*). This was also evident from the gene losses among different categories of CAZymes (carbohydrate-active enzymes), and SMCs (secondary metabolite clusters), as well as peptidases, transcription factors, and effectors. These patterns remained consistent when focusing solely on plant pathogens (*Figure 5—figure supplement 1*). The only gene classes that did not exhibit a reduction or showed gains in these three clades were tRNAs, genes involved in cytoskeleton functions, and cell motility (*Figure 5C*). Other IA species displayed more variability in changes to their gene repertoires, typically showing some gains in SMC genes (H2.2, H2.6). Two clades enriched with non-IA plant pathogens (H1, G) demonstrated the highest gains in host-related genes. Clade H1 experienced the most gene gains in genes related to secondary metabolite pathways, defense mechanisms, carbohydrate and amino acid transport and metabolism, whereas clade G experienced most gene gains in genes related to extracellular structures, secondary metabolite pathways, carbohydrate transport and metabolism, and the cell wall/membrane/envelope biogenesis.

Overall, we observed common gene gains, particularly in CAZy and SMC classes, among many crop (clades H1, G) and some tree pathogens (clade D, *Figure 5C*, *Figure 5—figure supplement 1*). On the other hand, insect-vectored species (clade H2.2), including plant pathogens (clades M1, H2.8, O), and specialized entomopathogens (H2.4, H2.6) exhibited variable levels of gene losses, sometimes in genes critical for host interactions (*Figure 5C*, *Figure 5—figure supplement 1*). An exception was the IA clade H2.3, which includes the genera *Metarhizium* and *Pochonia*, both known for their ability to adapt to various lifestyles or hosts.

## Discussion

Fungal pathogens are commonly linked to notably large genomes (*Raffaele and Kamoun, 2012*; *Wacker et al., 2023*), yet they exhibit considerable diversity in both genome size and structure (*Möller and Stukenbrock, 2017*). We focused on the class of Sordariomycetes, which includes fungi with diverse lifestyles, to investigate three questions: first, which genomic traits best explain genome size; second, whether genome size can be attributed to pathogenic lifestyle; and third, whether pathogens share common genomic traits. Our analysis revealed that gene number is the strongest predictor of

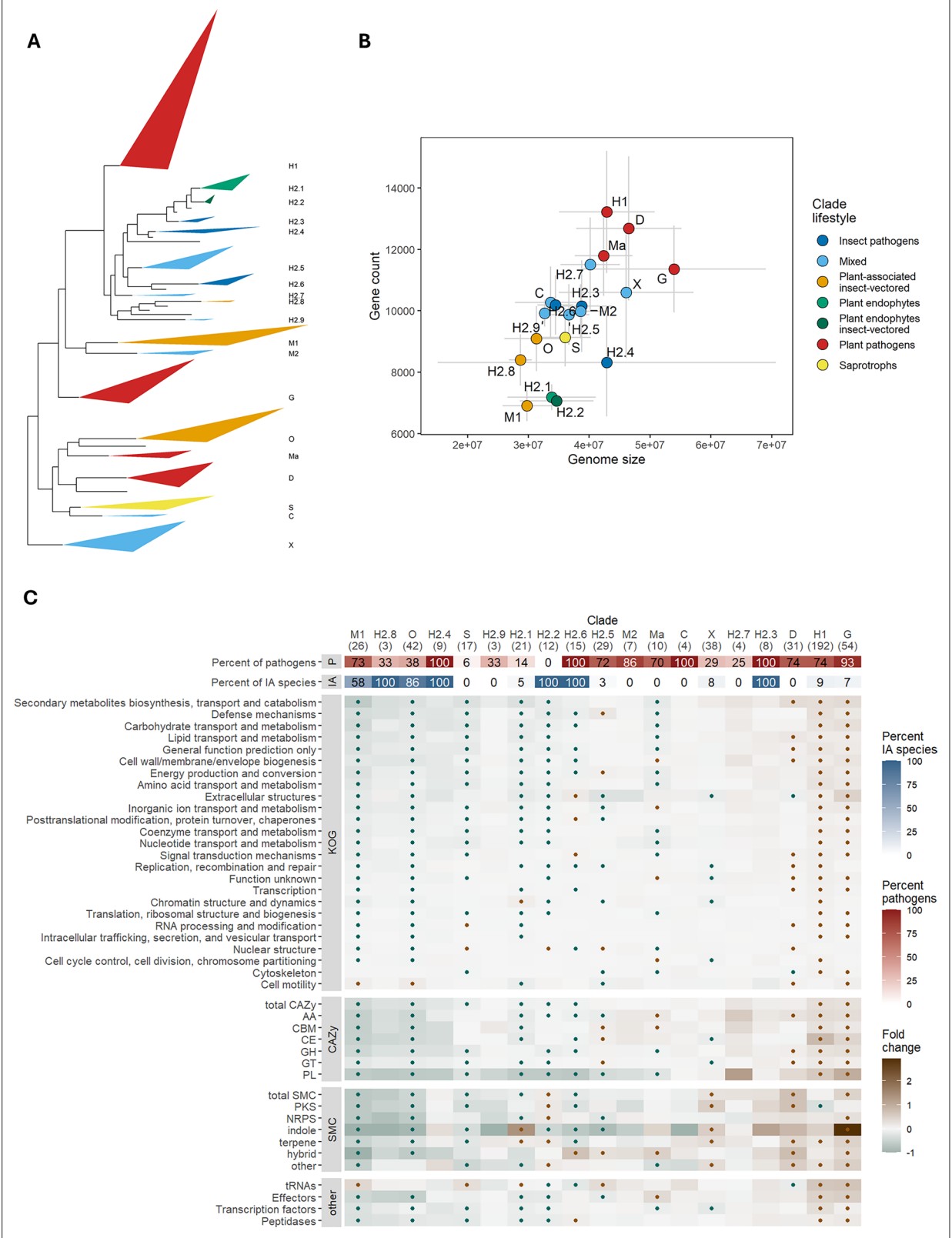

**Figure 5.** Insect-vectored clades lose genes involved in breaking plant host barriers. (**A**) Phylogenetic position of the selected clades. Names correspond to the ones in *Figure 2A* and colors correspond to general lifestyles explained in the legend in plot B. These broad categories indicate clades dominated by plant pathogenic fungi (H1, G, D), entomopathogens (H2.4, H2.6, H2.3), insect-vectored species (M1, O, H2.8, H2.2), saprotrophs (S), plant symbionts (H2.1, H2.2) or mixed lifestyle groups. The basis of triangles indicates span between minimum and maximum branch length for

*Figure 5 continued on next page*

*Figure 5 continued*

given clades. The height of the triangles was scaled with a factor of 0.2. Empty branches correspond to smaller clades whose members are not shown. (**B**) Means (dots) and standard deviations (error bars) of number of genes and genome size for selected clades. Sample sizes for each clade are given in panel C in parentheses under the clade identifier. (**C**) Heatmap shows the fold change of genes/clusters relative to the ancestral state ((observed - ancestral)/ ancestral state). Clades are shown in columns with the number of clade members in parentheses; functional classes are shown in rows. The dots indicate significant gain (brown) or loss (green) of genes/clusters across clade members estimated from 100 rounds of bootstrapping of 10 species in clades with ≥ 10 members. SMC: secondary metabolite clusters, CAZy: carbohydrate-active enzymes. The same heatmap but for pathogenic species only is shown in *Figure 5—figure supplement 1*. Data underlying figures are in *Figure 5—source data 1–4*.

The online version of this article includes the following source data and figure supplement(s) for figure 5:

**Source data 1.** Fold change of genes/clusters relative to the ancestral state.

**Source data 2.** Mean and high and low confidence intervals of fold change from bootstrapping species within clades.

**Source data 3.** Fold change of genes/clusters relative to the ancestral state in pathogenic species only.

**Source data 4.** Mean and high and low confidence intervals of fold change from bootstrapping species within clades in pathogenic species only.

**Figure supplement 1.** Insect-associated pathogens lose genes involved in breaking host barriers.

genome size, followed by non-coding regions. We found no correlation between genome size and pathogenicity. Instead, pathogenicity was on average best explained by the number of effectors and tRNAs, but the overall genome architecture of pathogens was strongly influenced by the type of interacting host and the presence of potential vector species.

## Genome size evolution in fungi

We found that the number of genes and their length—specifically, exon length and number—are the primary drivers of genome size in Sordariomycetes. While non-coding regions of the genome also contribute to this, they are less significant and are more frequently found in gene-poor species. These findings suggest that genes and non-coding regions do not evolve in concert or at the same rate.

To determine whether neutral processes can explain genome size in Sordariomycetes, we tested two hypotheses. The first posits that a decrease in effective population size ($N_e$) reduces the effectiveness of selection, including purifying selection, which in eukaryotes typically leads to a greater accumulation of deleterious non-coding elements and larger genome sizes (*Lynch and Conery, 2003*; *Lynch, 2007*). The second hypothesis suggests that, due to a higher rate of deletions compared to insertions, smaller $N_e$ results in more streamlined genomes, similar to observations in bacteria (*Kuo et al., 2009*; *Mira et al., 2001*). Using the genome-wide ratio of non-synonymous to synonymous substitution rates (dN/dS) as a proxy for $N_e$ (*Lefébure et al., 2017*; *Kelkar and Ochman, 2012*; *Marino et al., 2024*), we did not find the link between $N_e$ and the non-coding portion of the genome. Although we observed a weak negative correlation between the gene deletion rate and genome size, there was no correlation between the gene deletion rate and dN/dS.

The lack of a clear association of dN/dS with the non-coding content could arise from several factors. First, dN/dS may not be an appropriate indicator of $N_e$, or species analyzed may not be in mutation-selection balance. Second, fungi typically have much more gene-rich genomes than most other eukaryotes (*Wells and Feschotte, 2020*; *Merényi et al., 2023*), and the species in this study generally have modest genome sizes, which may weaken the relationship between $N_e$ and the non-coding regions. Lastly, the large non-coding content is often driven by the proliferation of transposable elements (TEs), which are known to be very dynamic. The absence of significant association between dN/dS and TE content on the long evolutionary timescales has been noted (*Marino et al., 2024*), suggesting possible variations in selective effects of TEs across different taxonomic groups.

Support for the neutral hypothesis of genome size evolution in our study is indirect. We found that species associated with insects—such as endoparasites and symbionts, therefore, species that likely undergo transmission bottlenecks and, in effect, have reduced $N_e$ (*Mira and Moran, 2002*)—have smaller genomes and in some cases, fewer genes. Additionally, insect-associated species have genes with fewer but longer exons, suggesting the evolution of a more streamlined gene architecture. While these findings align with neutral theories of genome evolution, further research is necessary to identify the primary processes, including any potential selective factors, that drive the dynamics of gene content in fungi.

## Genomic predictors of fungal pathogens

Large genome size driven by repeats has often been implicated as one of the hallmarks of fungal pathogens, especially fungal plant pathogens (*Raffaele and Kamoun, 2012*). However, in our analysis of over 500 genomes of Sordariomycetes, we found no evidence supporting a positive association between genome size or repeat content and pathogenic lifestyle. While a few species with the largest genomes are indeed plant pathogens and possess a high proportion of repeats, such instances appear to be exceptions rather than the rule on a broader scale. Although the high activity of transposable elements can contribute to genome expansion and facilitate the evolution of pathogenicity-related traits (*Hartmann et al., 2017*; *Wacker et al., 2023*), our findings suggest that this is not the predominant pathway for fungal pathogen evolution.

We found that the overall number of genes and genome size without repeats showed positive associations with pathogenicity on average. However, we can only speculate whether this pattern arises from drift or selection. Fungal pathogens comprise a diverse group of species, with specialists or generalists' adaptations and varying demographic histories. Pathogens vary from obligatory endoparasites, free-living saprotrophs that become opportunistic pathogens, to pathogens of common crops or plants with fluctuating or large $N_e$. Unfortunately, the specifics of the population dynamics and life histories of many pathogens remain largely unknown. Following the idea that smaller effective population sizes lead to smaller, more streamlined genomes in Sordariomycetes, the higher gene counts in pathogens may imply that most pathogens possess larger $N_e$ compared to non-pathogens. Alternatively, the greater number of genes in pathogens might reflect selective pressures favoring pathogen-specific gene expansions. An arms race between hosts and pathogens drives the continual evolution of new effector gene variants in pathogens, enabling them to evade the host immune system (*Boller and He, 2009*; *Stalder et al., 2023*). Our findings indicate the association of genes important for pathogenicity, such as effectors, with pathogens that can contribute to this overall pattern. In addition, we observe that clades with different lifestyles undergo distinct gene loss and gain dynamics, further suggesting a selective advantage for certain gene duplications.

Effectors and tRNAs were the only specific gene classes we examined in our analysis, and they proved to be more closely associated with pathogenicity than the overall gene count. The importance of effectors in pathogens is well established (*Stergiopoulos and de Wit, 2009*; *Wilson and McDowell, 2022*). For instance, mutations in effector genes can provide pathogens with an advantage by enabling them to evade host immune defenses (*Sánchez-Vallet et al., 2018*). The role of tRNA modifications in pathogenicity has also been recognized (*Hinsch et al., 2016*; *Li et al., 2023*), although the overall count of tRNAs remains less clear. In large eukaryotic genomes, short interspersed nuclear elements can lead to the misannotation of tRNAs (*Gibbs et al., 2004*). Therefore, the role of tRNAs in fungal pathogens warrants further investigation. Other gene families could be important predictors of pathogenicity. Carbohydrate-active enzyme profiles are reported to be different in pathogenic and non-pathogenic fungi (*Dort et al., 2023*; *Haridas et al., 2020*; *Hane et al., 2019*). Given that overall gene repertoires between pathogens can vary drastically, small-scale evolution of specialized genes or gene families can better reflect fungal transmissions to pathogenic lifestyles. Because of the uniform yet simplified gene annotation approach, the total number of genes may be underestimated in some assemblies in our dataset, as observed when comparing the same species in JGI MycoCosm. Although this pattern is not biased toward any particular group of species, access to high-quality, well-annotated genomes could provide a clearer picture of the relative contributions of specific gene families.

## Different genomic evolutionary pathways for fungal pathogens

Our study showed that the insect-association (IA) trait was negatively correlated with genome size and positively with exon length. Additionally, the total number of genes was also significantly reduced, particularly in insect-transmitted clades. Previous research showed that specialized fungi tend to have smaller genomes than generalist fungi (*Loos et al., 2024*), and the associations we observed between genomic traits and IA species may partially reflect the degree of specialization towards their hosts. In our dataset, IA species include several species of entomopathogens, many of which are host-specific. Plant pathogens vectored by insects are also largely host-specific. Unfortunately, it is challenging to obtain comprehensive data on the full range of host species represented in our dataset, which limits our ability to compare genome sizes between specialist and generalist species.

We found significant differences in gene evolution between IA and non-IA pathogens. Non-IA pathogens exhibited the highest gene evolution rates, whereas IA pathogens, on average, had the lowest. However, among the three clades of IA species included in our study, non-pathogenic species displayed a more pronounced reduction in effector repertoires compared to pathogenic species, highlighting the critical role of these genes across a broad range of fungal pathogens. Two IA clades, Ophiostomatales and Microascales, exhibited losses in gene classes important for host colonization and wood decomposition, such as CAZymes or secondary metabolite clusters (*Scharf et al., 2014*; *Cantarel et al., 2009*). This is likely because the insect vector often facilitates host entry in these species, rendering many genes redundant and prone to pseudogenization and loss. In the third clade, Hypocreales, patterns of gene losses and gains varied across species. Hypocreales is our dataset's largest order and includes both specialized and generalist species with diverse host and insect-associations. Furthermore, many species within this order have lifestyles that remain poorly understood. Distinguishing between pathogenic and non-pathogenic lifestyles can be challenging, particularly for species found on dead host tissues, which may function as opportunistic pathogens or secondary saprotrophs that benefit from decaying matter. These factors may partially contribute to the variation that we see across species.

## Conclusions

Our study shows that the genome size of Sordariomycetes is primarily driven by gene number, with non-coding regions contributing less. We found no association between pathogenicity and larger genome size or repeat content. Although many pathogenic species have a high number of genes, especially effectors and tRNAs, this pattern does not hold true for all species. While gene gains are common to some plant pathogens, others, such as insect-vectored pathogens exhibit gene losses, including in genes critical for host interactions. Insect-associated species also tend to have smaller genomes and longer exons, likely reflecting their level of specialization toward hosts and vectors— factors that may influence their effective population size. Overall, our findings suggest that the genome architecture of fungal pathogens is mainly shaped by the dynamics of pathogenicity-related genes and the nature of interactions with hosts and other species.

## Materials and methods
### Genome assemblies

We selected 14 strains (11 x *Ophiostoma* and 3 x *Leptographium*, *Supplementary file 1B*) for high-depth short-read Illumina sequencing. Isolates were grown on Malt Extract Agar medium (15 g l−1 agar, 30 g l−1 malt extract, and 5 g l−1 mycological peptone), and DNA was extracted with acetyl trimethylammonium bromide chloroform protocol. The fungal isolates were handled in a facility that has received a Plant Pest Containment Level 1 certification by the Canadian Food Inspection Agency. Library preparation and sequencing were conducted at the Génome Québec Innovation Center (Montréal, Canada). One strain (*O. quercus*) was sequenced with Illumina NovaSeq (paired-end 150 bp), and the rest with Illumina HighSeq X (paired-end 150 bp). Data quality was inspected with FastQC v0.11.2/8 (*Andrews, 2010*). Ten strains were used for de novo genome assembly (*Supplementary file 1B*). The depth of coverage of the generated data was between 32 x and 555 x. Reads were trimmed for adapters with Trimmomatic v0.33/0.36 (*Bolger et al., 2014*) using options 'ILLU-MINACLIP:adapters.fa:6:20:10 MINLEN:21' and overlapping reads were merged with bbmerge from BBTools v36/v37 (*Bushnell et al., 2017*). De novo genome assemblies were generated with SPAdes v3.9.1 (*Bankevich et al., 2012*) with options '-k 21,33,55,77,99 −−careful'. Mitochondrial DNA was searched in contigs with NOVOPlasty v3.8.3 (*Dierckxsens et al., 2017*) using the mitochondrial sequence of *O. novo-ulmi* as a bait (CM001753.1) and matching contigs were removed. Reads were remapped to the nuclear assembly with bwa mem v0.7.17 (*Li and Durbin, 2009*), and contigs with normalized mean coverage <5% and shorter than 1000 bp were also removed.

A total of 11 *Ophiostoma* strains were selected for long-read sequencing with PacBio (*Supplementary file 1B*). DNA extraction was done in the same way as for Illumina libraries, except that no vortexing and shaking were done to avoid DNA fragmentation. Library preparation and sequencing with the PacBio SMRTcell Sequel system were conducted at the Génome Québec Innovation Center (Montréal, Canada). The average depth of coverage of the generated data was

between 47 x and 269 x. De novo genome assemblies were generated with pb-falcon v2.2.0 (*Chin et al., 2016*). The range of parameters was tested and the final configuration files with the best-performing parameters for each species can be found with the submitted source code (https://github.com/aniafijarczyk/Fijarczyk_et_al_2025, copy archived at *Fijarczyk, 2025*). *O. quercus* was sequenced in two runs and read data from the two runs were combined together. *O. novo-ulmi* (H294) was sequenced in two runs, but only read data from one run was used for an assembly due to sufficient coverage. Assemblies were ordered according to *O. novo-ulmi* H327 genome (GCA_000317715.1) using Mauve snapshot-2015-02-13 (*Darling et al., 2004*). Assemblies were polished between two to four times by remapping long reads with minimap2 (pbmm v1 *Li, 2018*) and correcting assemblies using arrow (pbgcpp v1). We also performed one round of polishing with pilon v1.23 (*Walker et al., 2014*) after mapping short-read Illumina reads (bwa v0.7.17 *Li and Durbin, 2009*) to assemblies from this study (n=3) or from the previous study (*Hessenauer et al., 2020*) (n=7, *Supplementary file 1B*). The only exception was *O. quercus*, for which we had no corresponding short-read data; therefore, we performed four rounds of correction using arrow. The effectiveness of polishing was assessed by analyzing the completeness of genes with Busco v3 (*Waterhouse et al., 2018*). Mitochondrial genomes were assembled using Illumina assemblies as baits (or those of related species). Long reads were mapped to bait assembly with pbmm2 (*Li, 2018*), and a subsample of mapped reads was used for mtDNA assembly with mecat v2 (*Xiao et al., 2017*). Consecutive rounds of mapping and assembly were conducted until circularized assemblies were obtained. Nuclear assembly contigs with more than 50% of low-quality bases, those mapping to mtDNA, or with no mapping of Illumina reads (standardized mean coverage <5%) were filtered out. The genome assembly of *O. montium* was very fragmented and had a high percentage of missing conserved genes (74.3%), so instead of a PacBio assembly, an Illumina de novo assembly for this species was considered in further analysis.

In both short-read and long-read assemblies, contigs were searched for viral or bacterial contaminants by BLAST searches against bacterial or viral UniProt accessions and separately against fungal UniProt accessions. Contigs having more bacterial/viral hits in length than fungal hits were marked as contaminants and removed. Finally, repeats were identified with RepeatModeller v2.0.1 (*Flynn et al., 2020*) using the option -LTRStruct, and recovered repeat families together with fungal repeats from RepBase were used for masking assemblies with RepeatMasker v4.1.0 (*Tarailo-Graovac and Chen, 2009*).

Genes were annotated with Braker v2.1.2 (*Hoff et al., 2019*) using Augustus v.3.3.2 (*Stanke and Morgenstern, 2005*). To obtain *ab initio* gene models in *Ophiostoma novo-ulmi*, *ulmi*, *himal-ulmi*, *quercus*, and *triangulosporum*, a training file was generated using RNA-seq reads from *O. novo-ulmi* H327 (SRR1574322, SRR1574324, SRR2140676) mapped onto an *O. novo-ulmi* H294 genome with STAR v2.7.2b (*Dobin et al., 2013*). A training set of genes was generated with GeneMark-ES-ET v4.33 (*Lomsadze et al., 2005*). Final gene models were merged from *ab initio* models, models inferred from the alignment of RNA-seq reads (except *O. triangulosporum*) and *O. novo-ulmi* H327 proteins. The rest of the genome assemblies were annotated only *ab initio* using the Magnaporthe grisea species training file.

## Sordariomycetes genomes

A total of 580 genome assemblies (one reference genome per species) were downloaded from NCBI (accessed 21/05/2021), the two assemblies were downloaded from JGI MycoCosm (*S. kochii, T. guianense*), and two from other resources (*O. ulmi* W9 *Christendat, 2013*, *L. longiclavatum Wong et al., 2020*). The assemblies were obtained from NCBI rather than from other dedicated fungal genome resources, such as JGI MycoCosm, primarily because this dataset provided better representation of various taxonomic clades of interest, such as insect-associated species from the order Microascales. The dataset comprised assemblies produced with different sequencing technologies, which may affect their overall quality. However, both short-read and long-read-based assemblies were distributed in similar proportions across different lifestyles that we investigated (pathogenic vs. non-pathogenic, and insect-associated vs. non-insect-associated) and were present across phylogeny (Appendix 1). Indeed, long-read assemblies were on average longer and had a higher fraction of repeat content than short-read assemblies, but these differences did not affect the average differences between lifestyles that we investigated (Appendix 1).

Genes of all assemblies were subsequently annotated in a uniform manner. Assemblies were filtered to remove short contigs (<1000 bp) and contaminants, as described above. The 1000 bp threshold was chosen because it is a common cutoff used by many assemblers when producing contigs. Gene completion was assessed with Busco v3 (*Waterhouse et al., 2018*) using an orthologous gene set from Sordariomycetes. *Ab initio* gene models were obtained with Augustus v.3.3.2 (*Stanke and Morgenstern, 2005*) with different species-specific training files: Magnaporthe grisea, Fusarium, Neurospora, Verticillium longisporum, or Botrytis cinerea (*Supplementary file 1A*). Different species training sets were selected to reduce the phylogenetic distance between the annotated species and the training species. Examination of the impact of choice of the training species on estimated traits did not reveal potential influence (Appendix 1). Next, 31 assemblies were removed based on the low percentage of complete conserved genes (Busco score <85%) and one due to an excessive number of gene models (*Botrytis cinerea*). The number of genes was systematically underestimated (10% on average) compared to the gene numbers that had been submitted to NCBI for the corresponding species, but underestimation was not different between pathogenic or non-pathogenic fungi (Wilcoxon rank sum test, w=2994, *p*=0.57). We also observed that the number of genes was lower when compared with JGI MycoCosm assemblies of corresponding species. Again, this difference was consistent across studied lifestyles (Appendix 1).

## Phylogeny

The maximum likelihood tree was built from 1000 concatenated genes (retrieved with Busco v3 *Waterhouse et al., 2018*) with the highest species representation. Protein alignments were generated with mafft v7.453 (*Katoh and Standley, 2013*) with E-INS-i method (option '––genafpair ––ep 0 ––maxiterate 1000'), trimmed with trimal v1.4.rev22 (*Capella-Gutiérrez et al., 2009*) with option '-automated1' and converted into a matrix. The best protein evolution model, JTT +I + G4, was chosen as the most frequent model across all protein alignments, based on the BIC score in IQ-TREE v1.6.12 (*Nguyen et al., 2015*; *Kalyaanamoorthy et al., 2017*). The maximum likelihood tree was inferred with ultrafast bootstrap (*Hoang et al., 2018*), seed number 17629, and 1000 replicates implemented in IQ-TREE. Time-scaled phylogeny was inferred with program r8s v1.81 (*Sanderson, 2003*), setting the calibration point of 201 My at the split of *Neurospora crassa* and *Diaporthe ampelina*, retrieved from TimeTree (*Hedges and Kumar, 2005*).

## Species and trait selection

New and downloaded genome assemblies after filtering together amounted to 573. Few species were represented by more than one strain, in particular, multiple *Ophiostoma* strains sequenced in this study. To avoid potential biases related to the overrepresentation of some species, only one strain per species was retained, leaving 563 species in total, including 11 outgroup species.

Pathogenicity and insect-association traits were assigned based on a literature search. Pathogenicity was assigned if the species caused a well-recognized disease or pathogenicity was experimentally documented in at least one host species. We considered pathogens of plants, animals, and fungi, both obligatory and opportunistic. Insect-association included all types of relationships with insects, including pathogenic, symbiotic, and mutualistic. Insect-vectored species were limited to documented cases of insect transmission. Analyzed genomic traits included genome size (bp), number of genes, number of effectors, a fraction of repeat content, size of the assembly excluding repeat content (bp), GC content, the mean number of introns per gene, mean intron size (bp), mean exon size (bp), a fraction of genes with introns, mean intergenic length (bp), and number of tRNA and pseudo tRNA genes. Genome size is equivalent to assembly size after filtering. Genome size excluding repeat content is assembly size excluding regions masked by RepeatMasker (in this study or downloaded from NCBI). Repeat content was calculated as the proportion of masked bases in the raw assembly (after removing contaminated contigs, but before filtering for contig length) compared to the raw assembly size. Gene number corresponds to the number of all *ab initio* gene models obtained with Augustus. GC content is the proportion of GC bases in the total assembly. The mean number of introns per gene, intron and exon size, a fraction of genes with introns, and intergenic length were estimated from gff files with annotated gene models. tRNA and pseudo tRNA genes were obtained with tRNAscan-SE v2.0.9 (*Chan et al., 2021*). Effectors were identified from predicted proteins, first by running SignalP

v6 (*Teufel et al., 2022*) to identify proteins containing signal peptides, and then by running EffectorP v3 (*Sperschneider and Dodds, 2022*) on them to identify effectors.

## Correlation of genomic traits with genome size

The phylogenetic generalized least squares (PGLS) model of genome size was built with the gls function from the R package nlme v3-1.166 (*Pinheiro and Bates, 2025*). Maximum likelihood (ML) method was used, and the phylogenetic relationships were modeled with Pagel's lambda correlation structure. We considered 11 genomic traits (except for genome size and effectors) as explanatory variables. First, the correlations among pairs of genomic traits were calculated after transforming variables to independent contrasts using the pic function from the R package ape v5.8 (*Paradis et al., 2004*). Pairwise correlations were tested with non-parametric Spearman's rank correlation test. As it turned out, many variables were strongly correlated with each other. Therefore, instead of including all variables in the model, PCA was run with function prcomp() on 11 genomic variables to retain three main components, which explained 33%, 19%, and 13% of the trait variance, respectively. The loadings revealed that PC1 was mainly influenced by genic traits (exon length, introns, genome w/o repeats, genes), PC2 by non-coding traits (repeats, intergenic length), whereas PC3 by other traits (tRNA, genes with introns). We fitted the PGLS model with the three main PCs and interactions among them. The model was checked for homogeneity of variance and normal distribution of residuals and was refitted after removing 8 outliers. Model fit was assessed using the anova() function. To compute the importance of each component in the model, $R^2_{lik}$ was calculated with the R2() function from the rr2 v1.1.1 R package (*Ives, 2019*) after removing each component at a time and comparing it with the full model.

## Correlation between dN/dS and non-coding regions

Nonsynonymous to synonymous substitution rate (dN/dS) was calculated as a proxy for effective population size ($N_e$). Species with a larger $N_e$ are expected to exhibit a lower rate of nonsynonymous to synonymous substitution rate than species with smaller $N_e$ due to stronger constraints on nonsynonymous deleterious mutations. dN/dS was calculated with the codeml program in PAML v.4.10.6 (*Yang, 2007*), on a codon-wise alignment of 300 random concatenated orthologs identified with Busco. Specifically, for each species, calculations were performed on a subset of 10 most closely related species, with one species being the foreground branch and the rest serving as the background. Species, whose dN/dS estimate was based on a comparison with a recently diverged species showed elevated dN/dS values, likely because of the stronger contribution of segregating polymorphism compared to fixed differences to divergence estimates. To account for this time dependence, the logarithm of species divergence times was fitted to a logarithm of dN/dS using a generalized least square model, and model residues were taken to serve as dN/dS estimates in further correlations.

To test if species with smaller $N_e$ tend to accumulate more non-coding deleterious material, correlation between phylogenetically independent contrasts of time-corrected dN/dS estimates and log-transformed traits, such as repeat content, intergenic length, intron length, and number were calculated using Spearman's rank correlation.

## Correlation between the gene loss rate and genome size

To test if differences in deletion rate can account for genome size changes, a correlation between the rate of gene loss and genome size was tested. Orthologous gene families for 563 species were determined with OrthoFinder v2.5.2 (*Emms and Kelly, 2019*). Gene contractions were estimated by applying a birth-death model of gene evolution with CAFE v5 (*Mendes et al., 2021*). A single lambda for all branches was estimated, with a Poisson distribution for gene family counts at the root. As the likelihood scores could not be computed for all gene families, the dataset was limited to 527 gene families present at the root and with a maximum change of four genes across species. This procedure has likely led to the removal of many fast-evolving and big gene families and provides estimates on the rather conserved set of gene families. Lambda search was run three separate times to assure convergence. The rate of gene loss was calculated by summing gene family contractions and dividing by the calibrated branch length (in My). Correlation on phylogenetically independent contrasts of log-transformed genome size and log-transformed gene loss rate was calculated using Spearman's rank correlation.

## Association of genomic traits with pathogenicity and IA

Association of pathogenicity and IA with genomic traits was estimated using three approaches: (i) a reversible jump MCMC discrete model of evolution implemented in BayesTraits v3 (*Pagel and Meade, 2006*), (ii) a phylogenetic logistic regression with phyloglm() function in R package phylolm v2.6.5 (*Ho and Ané, 2014*), (iii) Random Forest classification using scikit-learn python v3 package. In tests of pathogenicity associations, analyses were also run separately within 10 subsets, where each subset comprised an equal number of pathogens and non-pathogens (n=190,190) sampled from five genome size bins (between 20 and 70 Mbp, with a 10 Mbp step).

BayesTraits tests the coevolution of a pair of binary traits by modeling eight possible transition rates between the two traits (transition of only one trait at a time). We modeled coevolution of each genomic trait with pathogenicity or IA. Genomic traits were converted into binary traits based on their median (0 if below median and 1 if above median), such that trait gain indicates increase of the size of that trait, and trait loss indicates decrease in the size of the trait. For each genomic trait, two models were compared, one with independent and one with dependent evolution of the two traits. In addition, to test if some transition rates are more frequent in one direction (gains vs losses of a trait), four models were run in which we set equal rates of change for the opposite transitions (for example, the rate of gain of genome size in pathogens was set to equal the rate of loss of genome size in pathogens), and compared them to dependent models using log Bayes factor. Based on all significant transition changes (gains or losses of one of the traits in the presence of the other), we determined if the genomic trait is positively associated with the pathogenicity/IA (gain of the trait for pathogens/IA, or loss of the trait in non-pathogens/non-IA). Finally, we also compared the dependent model with the covarion model in which coevolutionary transitions were allowed to vary across the phylogeny (*Venditti et al., 2011*). Models were selected if the log Bayes factor surpassed 4. Each model was run three times to check for consistency, for 21 mln iterations, 1 mln burn-in, and thinning of 1000.

Phylogenetic logistic regression was used to fit each genomic trait to pathogenicity/IA using the phyloglm() function in R with the 'logistic_MPLE' method, btol option (searching space limit) set to 30, and 1000 independent bootstrap replicates. Benjamini-Hochberg correction was applied to *p*-values. The scale of some genomic traits was adjusted. Assembly size with and without repeats was used in the unit of 10 Mbp, number of genes in units of 1000 genes, length of introns and intergenic length in the unit of 1 kb, length of exons in the unit of 100 bp, number of tRNAs, pseudo tRNAs, and effectors in the unit of 100 genes.

To train a machine learning classifier for predicting pathogens and IA species, we tested the accuracy of five classifiers (KNeighbors, SVC, DecisionTree, RandomForest, and GradientBoosting). In each case, randomized grid search and fivefold cross-validation was performed on the train data (80%), and balanced accuracy scores were obtained. Scores were averaged across 10 random splits into test and train datasets. Species with missing data were filtered out, and data were rescaled (between 0 and 1) for classification with SVC. As the Random Forest model showed one of the best scores, we used this model to train on the whole dataset and determine the most important features. Important features were determined using a mean decrease impurity method. In the final model, we used hyperparameters from the data split which gave the best balanced accuracy score. As the samples in the dataset are dependent through phylogenetic relationships, we also determined the impact of phylogeny on the model by including for each species distances to each node as additional features. We found that node distances never appeared among the top five most important features.

Ancestral reconstruction of genome size was performed using the fastAnc function in R package phytools v2.4–4 (*Revell, 2012*) and visualized on the tree with ggtree v3.15.0 R package (*Yu, 2020*) using the continuous option. To estimate lifestyle traits in selected ancestral nodes, a discrete model of evolution implemented in BayesTraits v3 was used with the same parameters as above.

To test the association of exon length and exon number between IA and non-IA clades, we obtained these features for 38 one-to-one orthologs found with OrthoFinder (described above). Mean values across IA species were compared to mean values in non-IA species for two clade pairs using a paired Mann-Whitney U test.

To determine whether long exon genes are more or less likely to be deleted compared to short exon genes, we examined the presence of gene families across different clade members. Gene families found in all clade members represent genes that are less likely to be deleted, whereas gene families present in only a few members of the clade represent genes that are more often deleted. We

then compared features of the common and rare gene families in terms of exon length and number, by fitting a negative binomial generalized linear model using the glm.nb() function from the MASS v7.3–61 package in R (*Venables and Ripley, 2002*).

## Evolution of pathogens with different lifestyles

The IA trait was modeled using phylogenetic logistic regression in the Phyloglm model, as described earlier, with each of the five genomic traits (genome size, exon length, number of genes, effectors, and tRNAs) tested individually, including pathogenicity as an interaction term. Models were fitted separately for three clades comprising species with a combination of pathogenicity and IA traits (H, M, and O/Ma/D/S).

To test if the rate of gene evolution varies between pathogenic species that are IA or non-IA, we used the birth-death model implemented in CAFE v5 to fit one, two, or four parameters of lambda on a subset of 527 small gene families. Pathogenicity and IA trait states were determined for each node in the phylogeny using function ace() in the R package phytools v2.4–4. A model with equal rates of transition was used for IA (ER model) and a model with non-equal transition rates was used for pathogenicity (ADR model). The birth-death models were compared using likelihood ratio tests.

## Functional gene classes

Functional gene annotations were determined using several databases. Protein sequences were searched against KOG (release 2003 *Tatusov et al., 2003*) and MEROPS Scan Sequences (release 12.1 *Rawlings et al., 2018*) using diamond v2.0.9 (*Buchfink et al., 2015*) with options '−−more-sensitive -e 1e-10' and against CAZymes (accessed 24/09/2021 *Cantarel et al., 2009*) with options '−−more-sensitive -e 1e-102'. Pfam domains were searched with pfam_scan.pl (*Mistry et al., 2021*) script with hmmer v3.2.1 (*Mistry et al., 2013*) and matched with Sordariomycetes transcription factors obtained from JGI MycoCosm database (accessed 9/12/2021). Secondary metabolite clusters were inferred from genome assemblies using antiSMASH v4.0.2 (*Blin et al., 2019*).

For each functional group of genes, and for each one of 19 selected clades, the ancestral number of genes/clusters was inferred in the most recent ancestor of the clade and its sister clade using the fastAnc function from R package phytools v2.4–4 (*Revell, 2012*). Fold change was calculated as a difference in observed and ancestral state divided by ancestral state and averaged across all clade members. Gains or losses of genes per clade were tested with a bootstrap of 10 species with 100 replicates for clades with ≥ 10 members.

## Acknowledgements

We thank Erika Dort for an access to lifestyle database to confirm pathogenicity traits of 90 species from our dataset. We thank Rohan Dandage, Ilga Porth, and Louis Bernier for comments and discussions about the manuscript. This project was funded by Genome Canada and Genome Québec BioSAFE project and a NSERC Discovery grant to CRL. CRL holds the Canada Research Chair in Cellular Systems and Synthetic Biology.

## Additional information

### Funding

| Funder | Grant reference number | Author |
| --- | --- | --- |
| Natural Sciences and Engineering Research Council of Canada | | Christian R Landry |
| Genome Canada | | Richard Hamelin |
| Génome Québec | | Richard Hamelin |

The funders had no role in study design, data collection and interpretation, or the decision to submit the work for publication.

## Author contributions
Anna Fijarczyk, Conceptualization, Data curation, Formal analysis, Validation, Investigation, Visualization, Methodology, Writing – original draft, Writing – review and editing; Pauline Hessenauer, Data curation, Writing – review and editing; Richard Hamelin, Supervision, Funding acquisition, Methodology, Writing – review and editing, Conceptualization; Christian R Landry, Resources, Supervision, Funding acquisition, Methodology, Writing – review and editing, Conceptualization

## Author ORCIDs
Anna Fijarczyk ⓘ https://orcid.org/0000-0003-2278-9842
Pauline Hessenauer ⓘ https://orcid.org/0009-0001-0689-5314
Richard Hamelin ⓘ https://orcid.org/0000-0003-4006-532X
Christian R Landry ⓘ https://orcid.org/0000-0003-3028-6866

Reviewer #1 (Public review): https://doi.org/10.7554/eLife.104975.4.sa1
Reviewer #2 (Public review): https://doi.org/10.7554/eLife.104975.4.sa2
Author response https://doi.org/10.7554/eLife.104975.4.sa3

# Additional files

## Supplementary files
Supplementary file 1. Supplementary tables. (A) Information on Sordariomycetes genome assemblies, including assembly identifiers, quality statistics and estimated trait values. (B) Sequencing information for species sequenced in this study. (C) Log marginal likelihoods and Bayes factors from likelihood ratio tests comparing different models of coevolution of pathogenicity with genomic traits. (D) Log marginal likelihoods and bayes factors from likelihood ratio tests comparing different models of coevolution of insect-association with genomic traits.

MDAR checklist

## Data availability
Raw short and long reads from sequenced genomes are available at NCBI under project number PRJNA841745. Genome assemblies have been deposited at GenBank under the accessions JANSLN000000000-JANSMH000000000. The versions described in this paper are versions JANSLN010000000-JANSMH010000000. The code used in this study, the phylogenetic tree, and configuration files for PacBio read assemblies are available on GitHub (https://github.com/aniafijarczyk/Fijarczyk_et_al_2025, copy archived at *Fijarczyk, 2025*) and Zenodo (doi: https://doi.org/10.5281/zenodo.17765751).

The following dataset was generated:

| Author(s) | Year | Dataset title | Dataset URL | Database and Identifier |
|---|---|---|---|---|
| Fijarczyk A, Hessenauer P, Hamelin RC, Landry CR | 2022 | Genome sequencing and assembly of dutch elm disease fungi and related species | https://www.ncbi.nlm.nih.gov/bioproject/PRJNA841745 | NCBI BioProject, PRJNA841745 |

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

## Appendix 1

### Impact of sequencing technology on annotation of trait values

We investigated the potential impact of sequencing technology used to generate genome assemblies on annotation of genomic features. For each genome assembly, information on sequencing technology was extracted using NCBI datasets v18.0.2. Genome assembly sequencing technology was classified as long-read if it contained technology, including Nanopore (ONT), PacBio, or Sanger sequencing; otherwise, it was classified as short-read assembly. 461 assemblies were classified as short-read, and 102 as long-read. Long-read assemblies were present across all of the large clades: Diaporthales (19%), Glomerellales (29%), Hypocreales (32%), Magnaporthales (10%), Microascales (12%), Ophiostomatales (29%), Sordariales (29%), Xylariales (61%). Pathogenic species comprised 63% of species in long-read vs 65% in short-read assemblies, and IA (insect-associated) species comprised 23% of species in long-read vs 25% in short-read assemblies (*Appendix 1—table 1*).

**Appendix 1—table 1.** Proportion of species with different lifestyles among short and long-read assemblies.

| Reads | Non-pathogen | Pathogen | Non-IA | IA |
|---|---|---|---|---|
| long | 0.37 | 0.63 | 0.77 | 0.23 |
| short | 0.35 | 0.65 | 0.75 | 0.25 |

The mean values of 13 traits were similar between long and short-read assemblies, with largely overlapping distributions (*Appendix 1—figure 1*). Since the two sets consist of non-overlapping species, variation around the mean is expected. Median values in long-read assembly species were higher for total assembly length by 3.16 Mb, repeat content by 0.04, mean intron length by 6 bp, and mean intergenic length by 518 bp, and they were lower in long-read assembly species for the number of genes by 548, effectors by 34, and tRNA genes by 13.

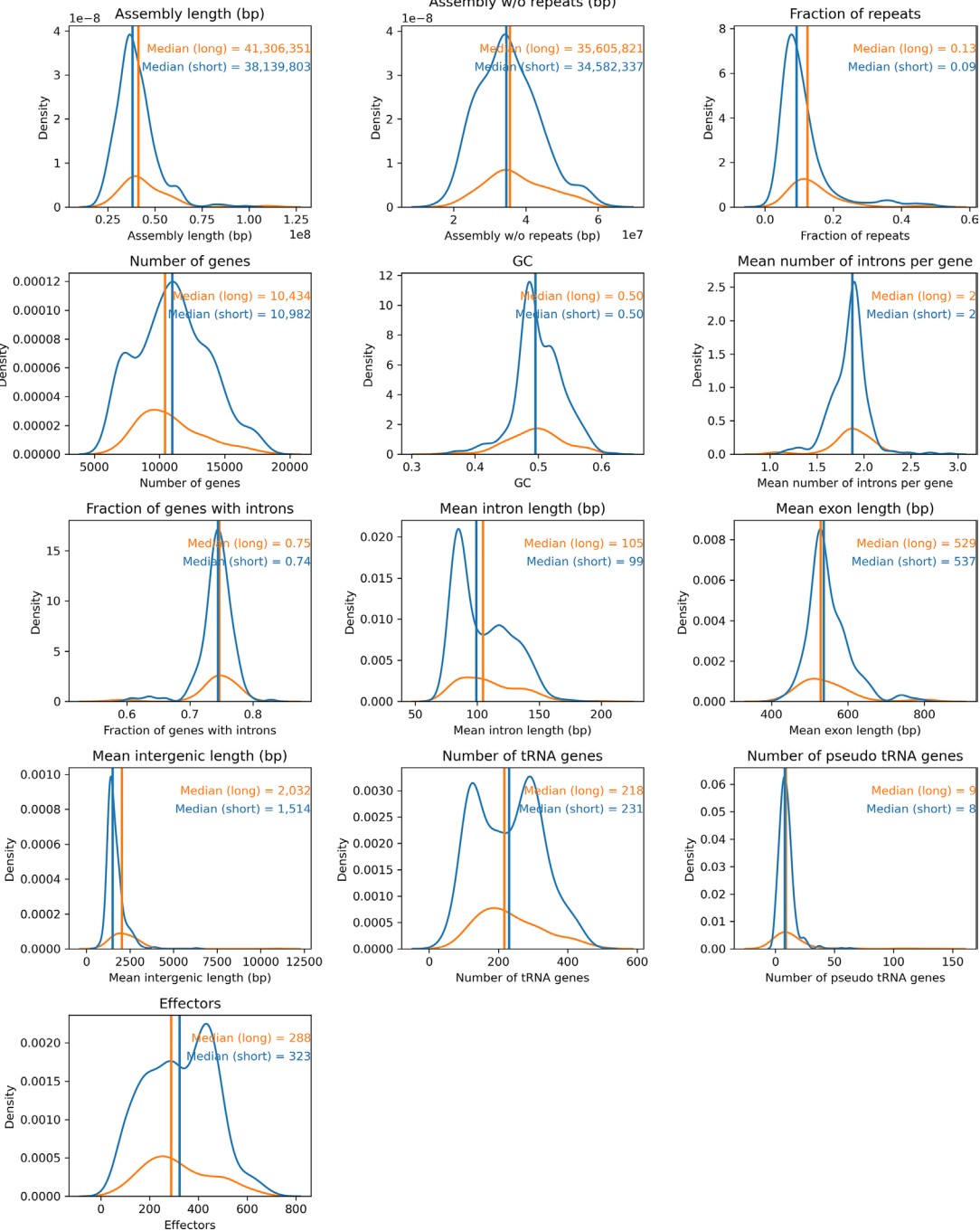

**Appendix 1—figure 1.** Distribution of trait values calculated for long-read assembly (orange) and short-read assembly (blue) species. Vertical lines indicate the position of median values.

To investigate if genomic traits vary between lifestyles in the same way for long and short-read assemblies, we compared trait distributions for long and short-read assemblies between pathogens and non-pathogens (*Appendix 1—figure 2*) and between IA and non-IA species (*Appendix 1—figure 3*). Differences in trait values between lifestyles were similar and in the same direction for both short and long-read assemblies, except for the mean intron length, and to a lesser degree in GC content and the number of pseudo tRNAs in pathogens vs. non-pathogens (*Appendix 1—figure 2*). Mean intron length was longer for pathogens than non-pathogens in long-read assemblies, but shorter for pathogens than non-pathogens in short-read assemblies.

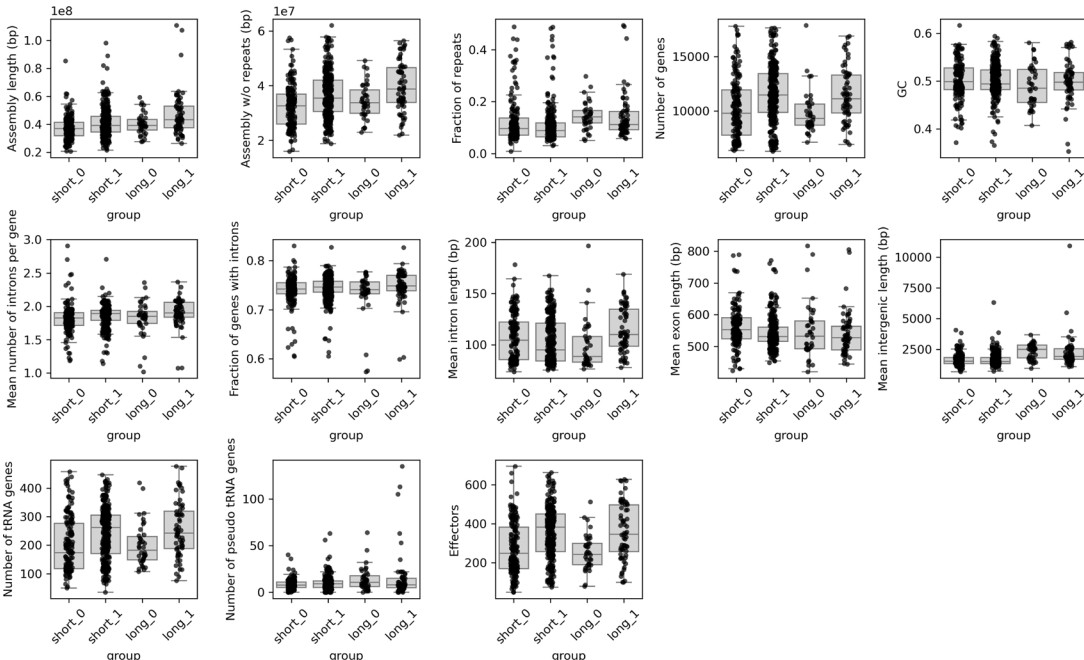

**Appendix 1—figure 2.** Distribution of genomic trait values for short and long-read assemblies separately for pathogenic (1) and non-pathogenic (0) species.

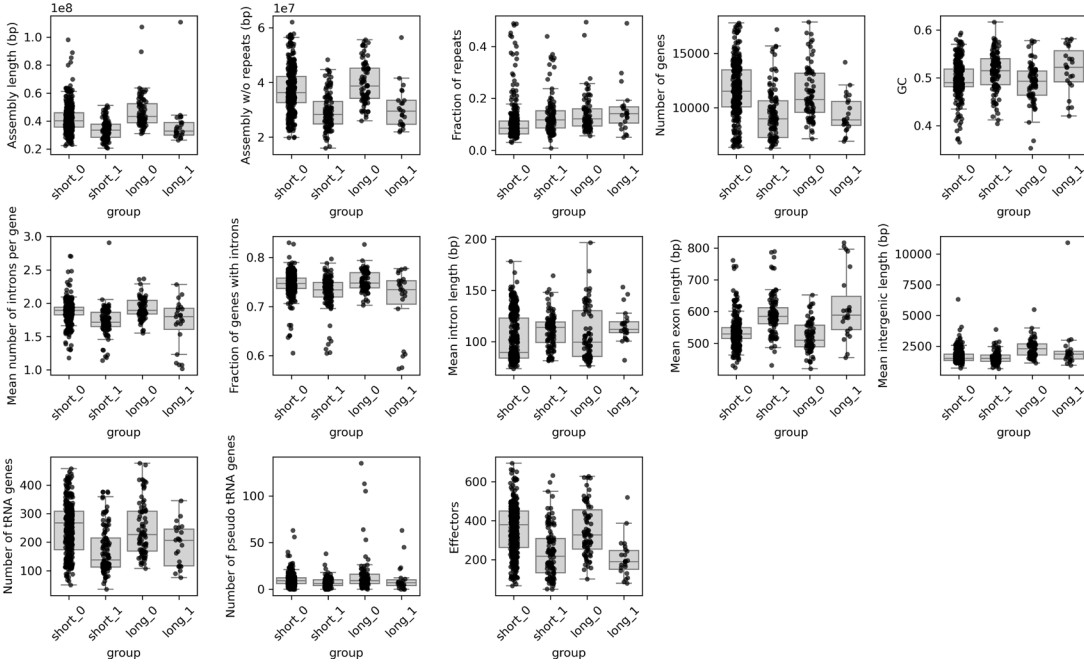

**Appendix 1—figure 3.** Distribution of genomic trait values for short and long-read assemblies separately for IA (insect-associated) (1) and non-IA (0) species.

## Comparison of assembly length and repeat content with JGI MycoCosm annotations

Genomic resources such as JGI MycoCosm provide another source of fungal genomes with standard pipelines for gene and repeat annotations. To investigate how well estimated genomic traits from our dataset (raw repeat annotation from NCBI and Augustus gene annotation) compare with those of JGI MycoCosm, we downloaded genome assemblies and gene annotations for published

Sordariomycetes species from JGI MycoCosm (downloaded on 17/05/2025). Only 58 species were matching our dataset, and 19 of them were also matching at the strain level. The species set (n=58) included 33 Hypocreales, 11 Xylariales, 5 Sordariales, 4 Glomerellales and single representatives of three other clades. The strain set (n=19) included 14 Hypocreales, 2 Xylariales, 2 Diaporthales, and 1 Coniochaetales species. Only three of these strains were sequenced with long-read sequencing technology (Sanger or PacBio). Genome assemblies were very similar in length between NCBI and MycoCosm datasets at the species level (median of 40.9 Mbp for NCBI vs 41.3 Mbp for MycoCosm), and at the strain level (median of 42.2 Mbp for NCBI vs 42.4 Mbp for MycoCosm, *Appendix 1—figure 4*). Differences between lifestyles were in the same direction for the NCBI and MycoCosm datasets (*Appendix 1—figure 4*). Repeat content (proportion of masked bases in the assembly) was higher for NCBI than MycoCosm assemblies both for the species (median of 0.091 for NCBI vs 0.053 for MycoCosm) and strain comparisons (median of 0.081 for NCBI vs 0.054 for MycoCosm, *Appendix 1—figure 4*). For the same strain comparison, repeat content in MycoCosm assemblies was lower for pathogens than non-pathogens (median of 0.054 in pathogens vs. 0.077 in non-pathogens) compared to NCBI assemblies which had similar repeat content for both lifestyles (median of 0.081 in pathogens vs. 0.076 in non-pathogens, *Appendix 1—figure 4*). Notably, this result is based on only four non-pathogenic species, which is too few to extrapolate to the entire dataset. The number of IA species in the dataset was generally very low (and absent from the same strain dataset) to draw meaningful conclusions.

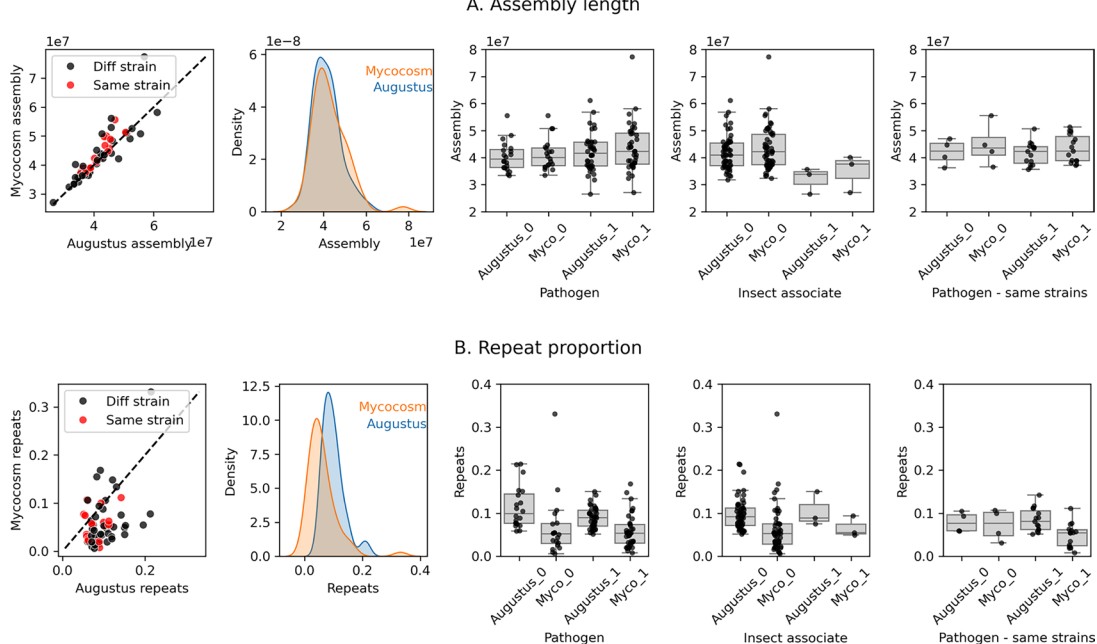

**Appendix 1—figure 4.** Comparison of assembly length and repeat annotations between matching species from our dataset (NCBI) and JGI MycoCosm. In box plots, categories '1' correspond to pathogens or IA (insect-associated) species, and '0' to non-pathogens or non-IA species.

## Comparison of gene annotations with JGI MycoCosm

Similarly, we compared five gene traits (number of genes, number of introns, intron length, exon length, and fraction of genes with introns) between our dataset annotated with Augustus and JGI MycoCosm annotations. For the JGI MycoCosm dataset, gene traits were calculated from gff files with the same scripts as those from Augustus. The number of genes was systematically higher for MycoCosm than Augustus datasets for almost all 58 matching species, and it was higher for both pathogenic and non-pathogenic species (*Appendix 1—figure 5*). Other gene traits were on average slightly lower for MycoCosm than Augustus datasets, in particular for the same strain comparisons in both pathogen and non-pathogenic species (*Appendix 1—figure 5*).

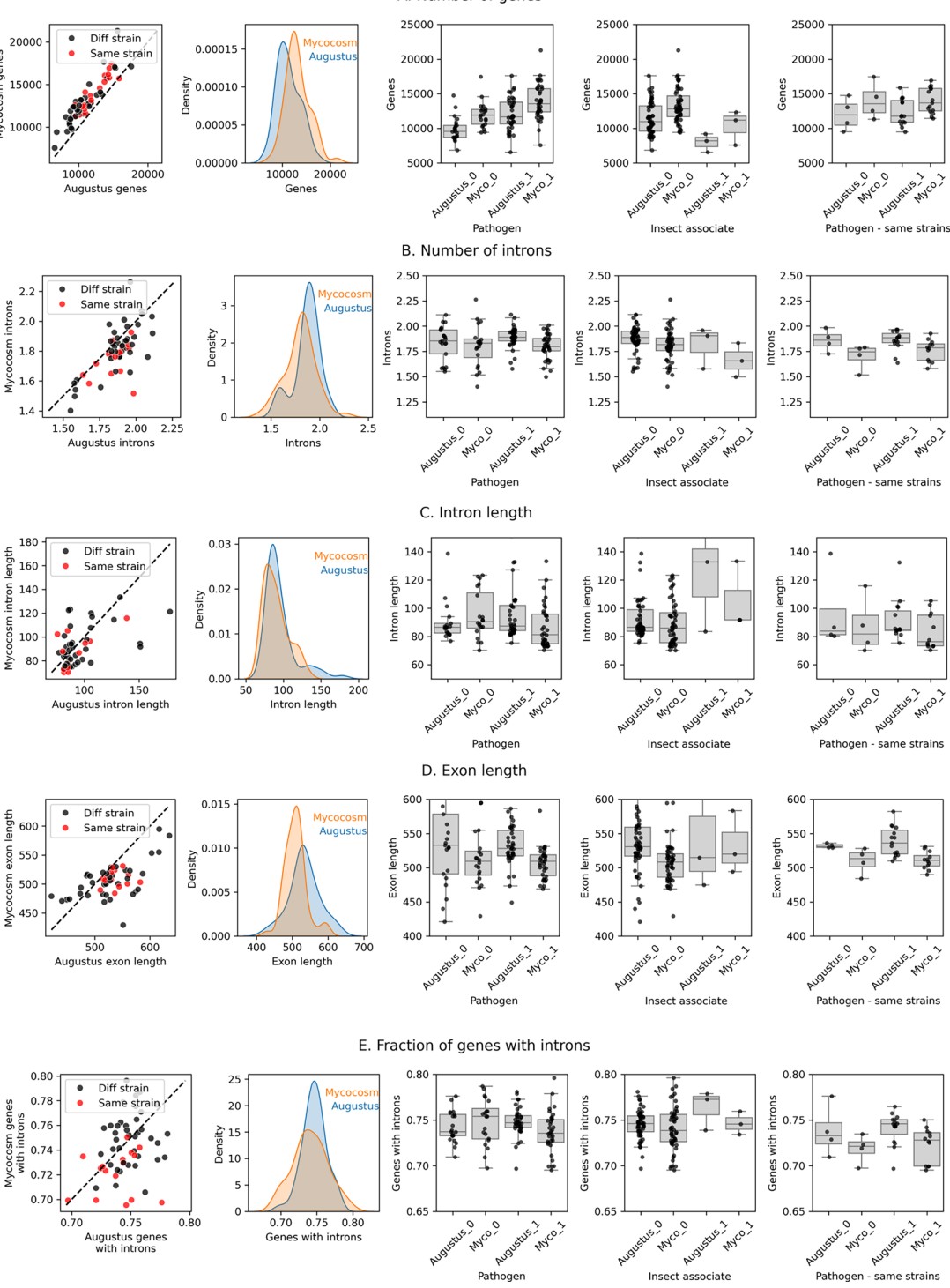

**Appendix 1—figure 5.** Comparison of gene annotations between matching species from our dataset (NCBI) and JGI MycoCosm. In box plots, categories '1' correspond to pathogens or IA (insect-associated) species, and '0' to non-pathogens or non-IA species.

## Impact of the training set on gene annotations

To investigate whether the choice of the species as a training set for Augustus gene annotation had an impact on gene traits, we compared five gene traits as a function of distance from the species used in a training set. Distance was calculated as the cumulative branch length distance estimated

from the phylogenetic tree using the dist.nodes() function from R package ape v5.8. The scatterplots are shown in *Appendix 1—figure 6*. Overall, there were no consistent patterns across training sets and within each training set differences between species belonging to different orders were usually more pronounced than those within orders. In the case of the Fusarium training set, there was a clear decrease in the number of genes with increasing distance, and an increase in intron and exon length with increasing distance (*Appendix 1—figure 6*). To verify if these trends could be impacted by a species selected for training, we compared gene traits between species annotated with Augustus (n=353) and species from the JGI MycoCosm dataset (n=146, including species not overlapping with Augustus) in three groups of species with increasing distance from Fusarium: species from the order Hypocreales belonging to the *Fusarium* genus (corresponding to clade H1 in this study), other species from the Hypocreales order (corresponding to clade H2 in this study) and species from the Microascales order. Even though representation for the Microascales was very small (n=2), and the mean values differed between Augustus and MycoCosm (as demonstrated above), the differences between three groups of species were of the same magnitude and direction (*Appendix 1—figure 7*), confirming that these trends are not driven predominantly by Augustus annotation.

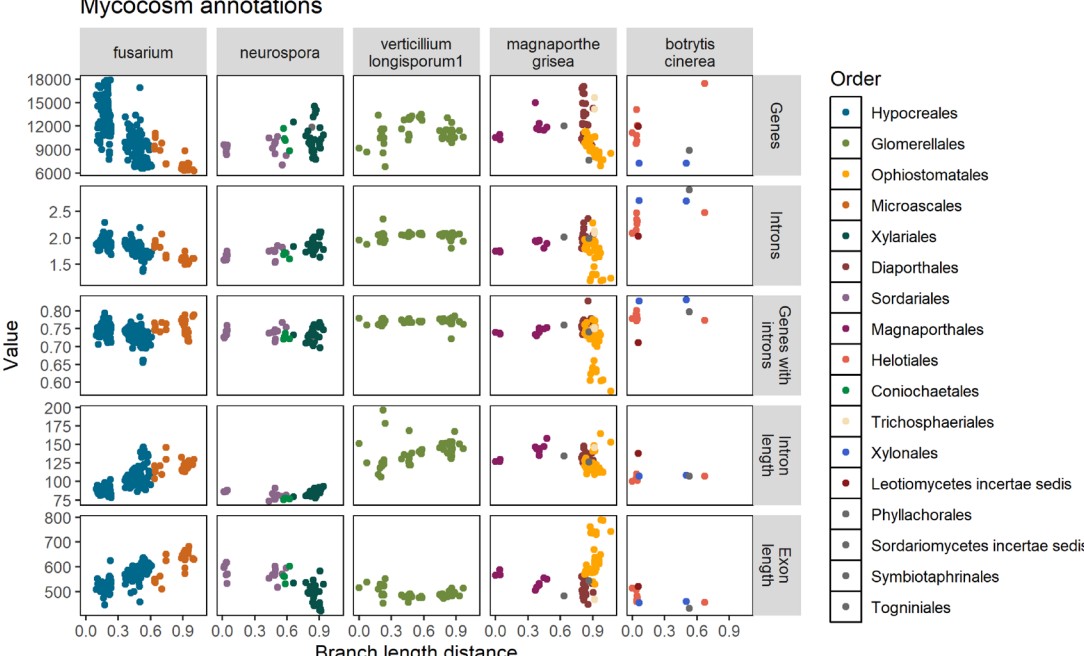

**Appendix 1—figure 6.** Gene trait values as a function of distance from the five species or genera used in Augustus training.

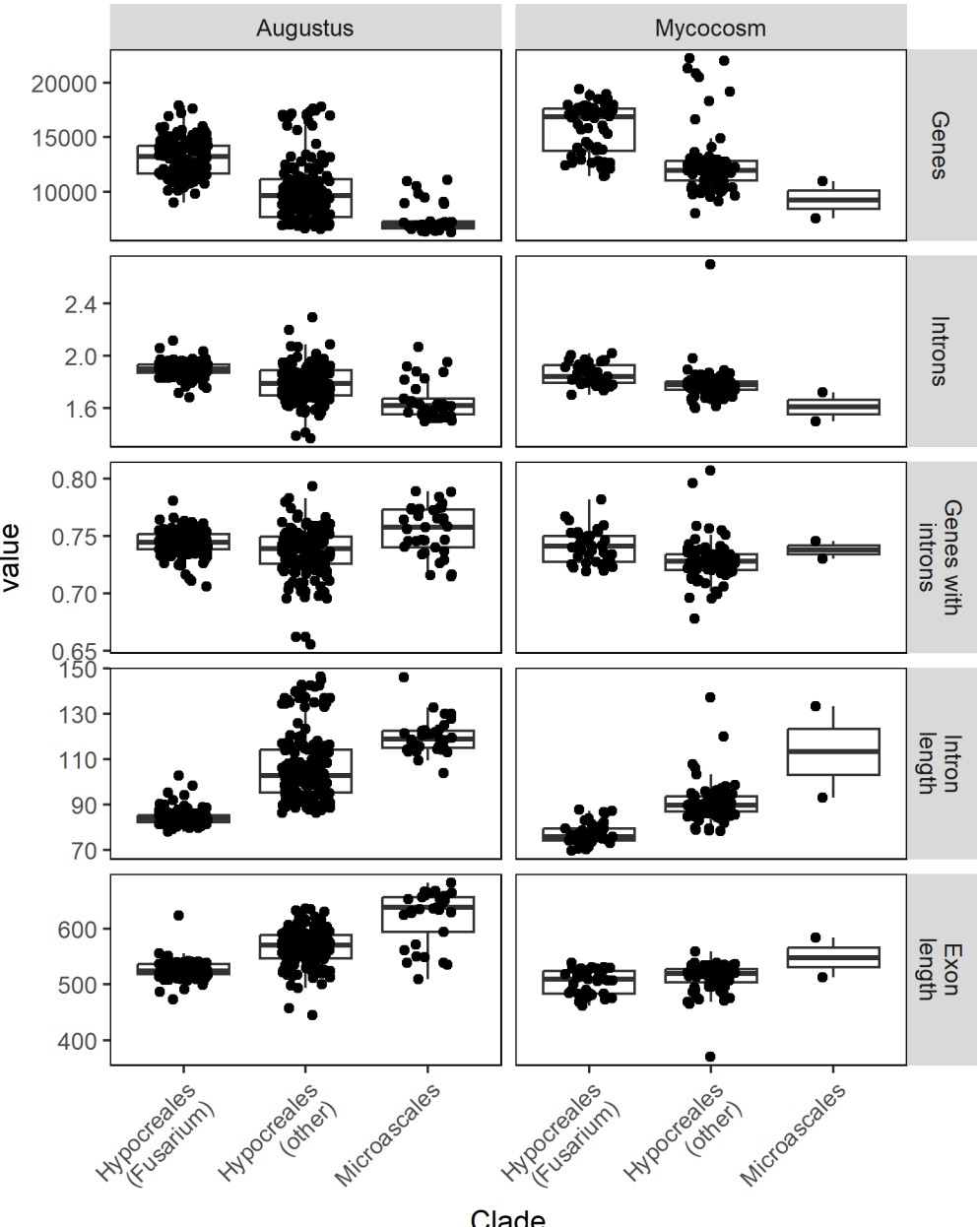

**Appendix 1—figure 7.** Distributions and medians of gene traits compared between all species from Hypocreales and Microascales retrieved from JGI MycoCosm, and all species from the same two orders in our dataset annotated with Augustus using 'fusarium' for training. Species were split into three groups with increasing distance from genus *Fusarium*, namely all *Fusarium* species in Hypocreales order, all other Hypocreales, and Microascales.

