## [Editor Report · eLife Assessment]

This **important** study addresses a topic that is frequently discussed in the literature but is under-assessed, namely correlations among genome size, repeat content, and pathogenicity in fungi. Contrary to previous assertions, the authors found that repeat content is not associated with pathogenicity. Rather, pathogenic lifestyle was found to be better explained by the number of protein-coding genes, with other genomic features associated with insect association status. The results are considered **solid**, although there remain concerns about potential biases stemming from the underlying data quality of the analyzed genomes.

---

## [Referee Report · Reviewer #1 (Public review)]

Summary:

The manuscript "Lifestyles shape genome size and gene content in fungal pathogens" by Fijarczyk et al. presents a comprehensive analyses of a large dataset of fungal genomes to investigate what genomic features correlate with pathogenicity and insect associations. The authors focus on a single class of fungi, due to the diversity of life styles and availability of genomes. They analyze a set of 12 genomic features for correlations with either pathogenicity or insect association and find that, contrary to previous assertions, repeat content does not associate with pathogenicity. They discover that the number of protein coding genes, including total size of non-repetitive DNA does correlate with pathogenicity. However, unique features are associated to insect associations. This work represents an important contribution to the attempts to understand what features of genomic architecture impact the evolution of pathogenicity in fungi.

Strengths:

The statistical methods appear to be properly employed and analyses thoroughly conducted. The size of the dataset is impressive and likely makes the conclusions robust. The manuscript is well written and the information, while dense, is generally presented in a clear manner.

---

## [Referee Report · Reviewer #2 (Public review)]

Summary:

In this paper, the authors report on the genomic correlates of the transition to the pathogenic lifestyle in Sordariomycetes. The pathogenic lifestyle was found to be better explained by the number of genes, and in particular effectors and tRNAs, but this was modulated by the type of interacting host (insect or not insect) and the ability to be vectored by insects.

Strengths:

The main strengths of this study lie in (i) the size of the dataset, and the potentially high number of lifestyle transitions in Sordariomycetes, (ii) the quality of the analyses and the quality of the presentation of the results, (iii) the importance of the authors' findings.

Weaknesses:

The weakness is a common issue in most comparative genomics studies in fungi, but it remains important and valid to highlight it. Defining lifestyles is complex because many fungi go through different lifestyles during their life cycles (for instance, symbiotic phases interspersed with saprotrophic phases). In many fungi, the lifestyle referenced in the literature is merely the sampling substrate (such as wood or dung), which does not necessarily mean that this substrate is a key part of the life cycle. The authors discuss this issue, but they do not eliminate the underlying uncertainties.

[Editors' note: this version was assessed by the editors, without involving the reviewers again.]

---

## [Author Response]

The following is the authors’ response to the previous reviews

**Reviewer #1 (Recommendations for the authors):**
I think the authors did a fantastic job investigating the annotation issues I brought up in the first round. I am somewhat assured that the size of the dataset has prevented any real systematic issues from impacting their results. However, there are many clear underlying biases in the data, as the authors show, which could have a number of unexpected impacts on the results. For example, the consistently lower gene numbers could be biased towards certain types of genes or in certain lineages, making the CAZyme analysis unreliable. I do not agree with the author's choice to put these results in as a supplement with little or no other references to it in the main manuscript. Many of the conclusions that are drawn should be hedged by these findings. There should at least be a rational given for why the authors took the approach they did, such as mentioning the points they brought up in the response.

We thank the reviewer for the positive assessment of our revision. We added text in the Discussion acknowledging limitations of the gene annotation approach.

“Because of the uniform yet simplified gene annotation approach, the total number of genes may be underestimated in some assemblies in our dataset, as observed when comparing the same species in JGI Mycocosm. Although this pattern is not biased toward any particular group of species, access to high-quality, well-annotated genomes could provide a clearer picture of the relative contributions of specific gene families.”

We also added more text in the Methods (section "Sordariomycetes genomes") mentioning in more detail the investigation of potential biases related to assembly quality and annotation (with reference to Supplementary Results).

A couple minor corrections:Figure 1C, both axes say PC1?

Fixed.

Figure S12, scales don't match so it's hard to compare, axis labels are inconsistent.

Fixed.

**Reviewer #2 (Recommendations for the authors):**
I congratulate the authors on the revision work. Their manuscript is very interesting and reads very well.I found several occurrences of « saprophyte ». Note that « saprotoph » is much better since fungi are not « phytes ».

We thank the reviewer for positive feedback. The occurrences of “saprophytes” were corrected.